# Complementary networks of cortical somatostatin interneurons enforce layer specific control

Alexander Naka[1], Julia Veit[1], Ben Shababo[1], Rebecca K Chance[2], Davide Risso[1,2,3,4], David Stafford[2], Benjamin Snyder[2], Andrew Egladyous[2], Desiree Chu[2], Savitha Sridharan[2], Daniel P Mossing[5], Liam Paninski[6,7,8,9], John Ngai[1,2,10], Hillel Adesnik[1,2]*

[1]Helen Wills Neuroscience Institute, University of California, Berkeley, Berkeley, United States; [2]Department of Molecular and Cell Biology, University of California, Berkeley, Berkeley, United States; [3]Department of Statistical Sciences, University of Padova, Padova, Italy; [4]Division of Biostatistics and Epidemiology, Department of Healthcare Policy and Research, Weill Cornell Medicine, New York, United States; [5]Department of Biophysics, University of California, Berkeley, Berkeley, United States; [6]Neurobiology and Behavior Program, Columbia University, New York, United States; [7]Center for Theoretical Neuroscience, Columbia University, New York, United States; [8]Departments of Statistics and Neuroscience, Columbia University, New York, United States; [9]Grossman Center for the Statistics of Mind, Columbia University, New York, United States; [10]QB3 Functional Genomics Laboratory, University of California, Berkeley, Berkeley, United States

**Abstract** The neocortex is functionally organized into layers. Layer four receives the densest bottom up sensory inputs, while layers 2/3 and 5 receive top down inputs that may convey predictive information. A subset of cortical somatostatin (SST) neurons, the Martinotti cells, gate top down input by inhibiting the apical dendrites of pyramidal cells in layers 2/3 and 5, but it is unknown whether an analogous inhibitory mechanism controls activity in layer 4. Using high precision circuit mapping, in vivo optogenetic perturbations, and single cell transcriptional profiling, we reveal complementary circuits in the mouse barrel cortex involving genetically distinct SST subtypes that specifically and reciprocally interconnect with excitatory cells in different layers: Martinotti cells connect with layers 2/3 and 5, whereas non-Martinotti cells connect with layer 4. By enforcing layer-specific inhibition, these parallel SST subnetworks could independently regulate the balance between bottom up and top down input.
DOI: https://doi.org/10.7554/eLife.43696.001

*For correspondence:
hadesnik@berkeley.edu

Competing interests: The authors declare that no competing interests exist.

## Introduction

The neocortex is divided across its vertical axis into discrete layers. Excitatory principal cells (PCs) in each layer differentially encode and process sensory information (*Feldmeyer, 2012*; *Harris and Shepherd, 2015*), due in part to the fact that they receive different external inputs. 'Bottom-up' sensory inputs primarily enter L4, while 'top down' inputs target PCs in supragranular and infragranular layers. The relative impact of these two main input pathways is likely to be crucial for sensory-guided behavior. Dendrite-targeting somatostatin (SST)-expressing interneurons are well known for shaping sensory coding through lateral and recurrent inhibition, and for gating top-down input by regulating dendritic spiking and synaptic plasticity (*Adesnik et al., 2012*; *Kim, 2016*; *Kvitsiani et al., 2013*;

*Makino and Komiyama, 2015*; *Urban-Ciecko and Barth, 2016*; *Kato et al., 2017*; *Veit et al., 2017*; *Adesnik, 2017*). While it is well established that SST neurons send and receive synaptic connections with neurons across multiple cortical layers (*Apicella et al., 2012*; *Jiang et al., 2015*; *Jiang et al., 2013*; *Anastasiades et al., 2016*; *Otsuka and Kawaguchi, 2009*; *Yoshimura and Callaway, 2005*; *Kapfer et al., 2007*), it is unclear whether SST circuits impact all layers in a global manner, or if instead they selectively control specific layers, which could have profound impacts for cortical computation.

Studies which have examined connectivity of SST cells (*Fino and Yuste, 2011*) and GABAergic interneurons more generally (*Bock et al., 2011*; *Scholl et al., 2015*; *Hofer et al., 2011*; *Packer and Yuste, 2011*) have found that inhibitory neurons interconnect with excitatory cells densely and non-selectively, which has led to the emerging hypothesis that excitatory circuits are overlaid by a 'blanket of inhibition'. While these studies have usually focused on connectivity within a single layer, one possibility is that this principle generalizes to the rest of the circuit, meaning that SST cells wire up irrespective of layer to globally regulate cortical networks. However, the SST population is highly heterogeneous (*Urban-Ciecko and Barth, 2016*; *Yavorska and Wehr, 2016*) and can be divided into multiple subgroups of cells which exhibit distinct electrophysiological, morphological, genetic (*Tasic et al., 2016*), and in vivo functional properties (*Kim, 2016*; *Kvitsiani et al., 2013*; *Muñoz et al., 2017*; *Ma et al., 2010*; *Reimer et al., 2014*; *Nakajima et al., 2014*). Intriguingly, some of these subgroups target their axons to different laminar domains (*Ma et al., 2006*; *Muñoz et al., 2014*; *Nigro et al., 2018*), and it has been hypothesized that subgroups of SST cells might be specialized to differentially modulate the activity of specific layers (*Muñoz et al., 2017*), perhaps by forming distinct subnetworks. Such an architecture would allow for independent gating of different cortical pathways by complementary networks of SST interneurons.

We addressed these contrasting hypotheses by combining high-resolution optogenetic circuit mapping, paired intracellular recordings, single-cell RNA sequencing, and in vivo optogenetics. With a focus on SST neurons in layer 5, the layer in which they are most numerous, we identified two distinct sub-networks of SST neurons with strikingly contrasting connectivity and in vivo function. The first sub-group of SST neurons was composed of Martinotti cells (MCs), a well-studied cell type which has classically been defined by its ascending axonal projection to L1. The second subgroup was morphologically, transcriptionally, and synaptically distinct from MCs and composed of SST neurons that primarily target L4 instead of L1. The connectivity and function of this second sub-class of SST neurons, which have sometimes been referred to simply as 'non-Martinotti' cells or 'NMCs' (*Muñoz et al., 2017*; *Ma et al., 2006*; *Nigro et al., 2018*), are largely unknown.

Our data on L5 SST neurons show that these two SST subgroups comprise highly distinct inhibitory subnetworks that exhibit exquisitely specific and strikingly complementary laminar patterns of connectivity and in vivo impact. MCs receive input from L2/3 and L5, whereas NMCs receive input from L4 and the L5B/L6 border. In turn, MCs provide reciprocal inhibition to PCs in L5 but not to those in L4, while NMCs selectively inhibit PCs in L4. Optogenetically activating MCs and NMCs in vivo results in extremely distinct laminar patterns of suppression, suggesting they may have contrasting roles in sensory computation and behavior. Single-cell RNA sequencing on >2000 individual SST neurons revealed transcriptomically defined SST sub-classes that showed distinct somal lamination profiles across the cortical range of depth. Taken together, these results demonstrate that of these, two major subgroups of cortical SST cells, by virtue of their layer-selective synaptic connectivity, can independently modulate the activity of different cortical layers during sensation. This highly selective synaptic and functional architecture supports a model in which distinct sub-networks of SST neurons may fine tune the balance of activity across the layers of the neocortex.

## Results

### Two distinct sub-networks of SST neurons defined by layer-specific connectivity

To probe the synaptic architecture of SST circuits, we employed a combination of one and two photon optogenetics, single cell reconstructions, and paired recording. To make targeted recordings from SST neurons belonging to putatively different sub-classes, we took advantage of transgenic reporter mouse lines that label either all SST neurons (*Sst*-IRES-Cre) (*Taniguchi et al., 2011*), or

different anatomical sub-classes of SST neurons in the barrel cortex (GIN, X94, and X98) (*Ma et al., 2006*; *Oliva et al., 2000*). We focused our investigation on SST neurons in L5 which harbors a large and diverse population of SST cells (*Markram et al., 2004*; *Rudy et al., 2011*). Consistent with prior

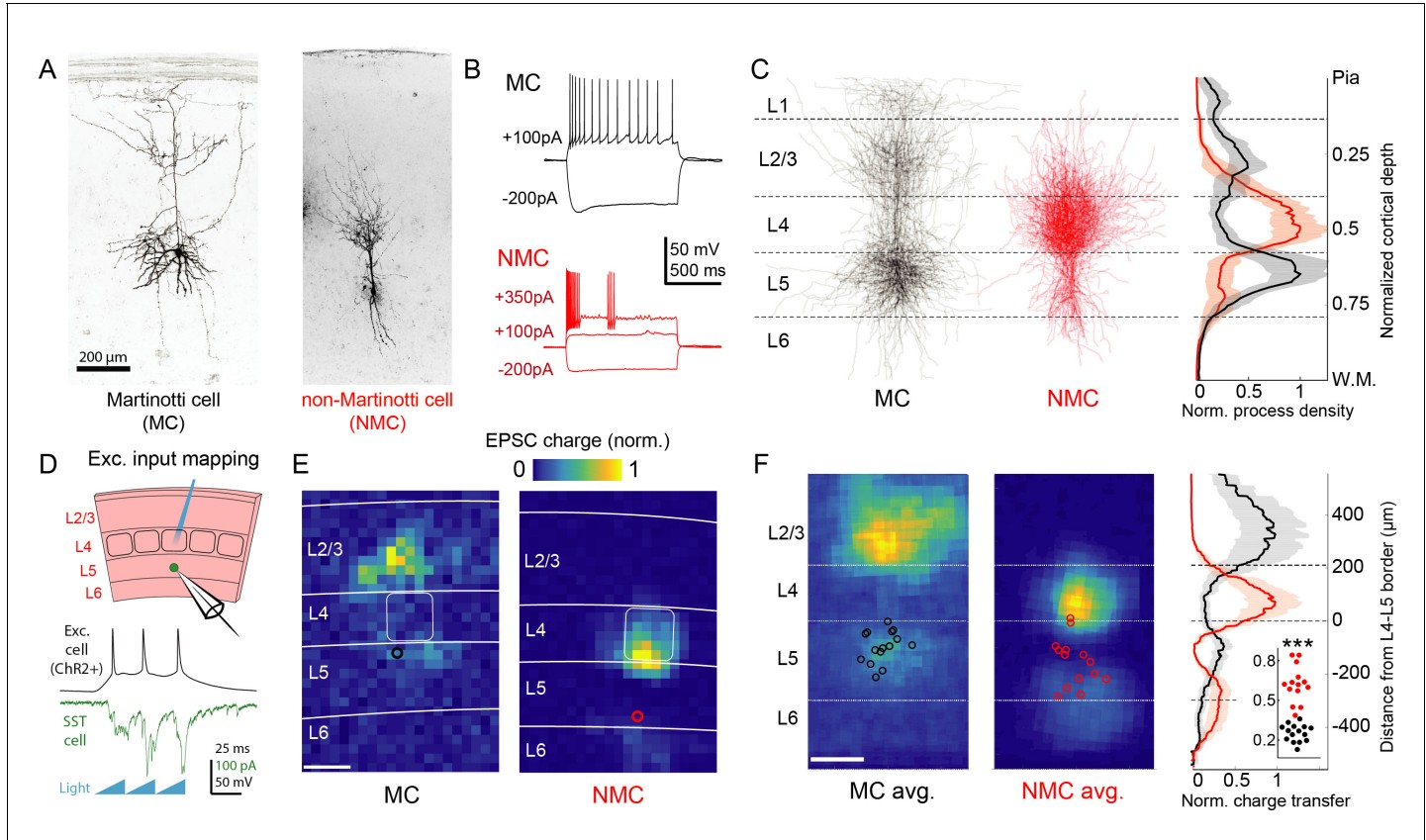

**Figure 1.** Optogenetic circuit mapping reveals complementary synaptic input patterns to two subtypes of L5 SST cells. (**A**) Confocal images of dye filled neurons revealing two morphological phenotypes of L5 SST cells. Left: an L5 GIN cell. Right: an L5 × 94 cell. Scale bar: 200 µm. (**B**) Example traces during current step injections from an L5 GIN cell (black) and an L5 × 94 cell (red). (**C**) Left: Overlaid morphological reconstructions of L5 GIN/MC cells (black, n = 14) and L5 × 94/NMCs (red, n = 10) showing differences in laminar distribution of neurites. Right: Normalized neurite density versus cortical depth for L5 GIN (black) and L5 × 94 cells (red). Data are represented as mean ±C.I. Note that these reconstructions do not distinguish between axon and dendrite; for detailed morphological analysis of these cells, see (*Ma et al., 2006*) (**D**) Schematic of experimental configuration. A digital micromirror device was used to focally photo-stimulate excitatory cells in different regions of the slice in order to map the spatial profile of excitatory inputs to GFP +L5 MCs (Emx1-Cre; GIN) or GFP +L5 NMCs (Emx1-Cre; X94). (**E**) Example heat maps of median EPSC charge transfer evoked at each stimulus site for example L5 SST cells. Left: An L5 MC that received inputs from L5 and L2/3. Right: An L5 NMC that received inputs from L4 and the L5/6 border. Soma locations are indicated by red/black bordered white dot). Scale bar: 200 µm. (**F**) Left: Grand averages of input maps reveal cell-type specific patterns of laminar input. Soma locations are indicated as above. Right: Normalized charge transfer versus distance from L4-L5 border for MC (black) and NMC (red) populations. Scale bar: 200 µm. Inset: Swarm plots showing the proportion of total evoked charge transfer in each map that originated from sites in L4 +L6, that is [L4+L6] / [L2/3+L4+L5+L6] for the MC (black; median, 27%; range, 13–36%) and NMC (red; median, 62%; range, 38–84%) populations. Proportions were significantly different between L5 MCs and L5 NMCs (25 ± 3% in n = 15 MCs versus 62 ± 7% in n = 14 NMCs, mean ±C.I.; p=6.5 · 10$^{-10}$; two-sample t-test). See also *Figure 1—figure supplement 1–4*.

DOI: https://doi.org/10.7554/eLife.43696.002

The following figure supplements are available for figure 1:

**Figure supplement 1.** Intrinsic properties of L5 SST cells.
DOI: https://doi.org/10.7554/eLife.43696.003

**Figure supplement 2.** Morphological reconstruction of L5 SST cells.
DOI: https://doi.org/10.7554/eLife.43696.004

**Figure supplement 3.** Comparison of intrinsic properties and excitatory inputs for L4 and L5 SST cells.
DOI: https://doi.org/10.7554/eLife.43696.005

**Figure supplement 4.** Excitation profiles of ChR2 +cells in Emx-Cre DMD-based one photon optogenetic mapping experiments.
DOI: https://doi.org/10.7554/eLife.43696.006

data, the anatomy and intrinsic physiology of L5 GIN cells demonstrated that they are MCs (41/44; 93%; *Figure 1a* and *Figure 1—figure supplement 1*) (*Wang et al., 2004*; *Kawaguchi, 1993*; *Fanselow et al., 2008*; *Kinnischtzke et al., 2012*). By similar analyses, neurons in the X94 line were non-Martinotti cells (32/35; 91%), which formed a dense axonal plexus in L4 rather than L1 (*Figure 1a*, *Figure 1—figure supplement 2*) and exhibited quasi-fast-spiking electrophysiological phenotypes (*Figure 1b*, *Figure 1—figure supplement 1*). While the X94 line labels only ~15% of SST cells in L5 (*Ma et al., 2006*), additional recordings from L5 SST-TdT cells suggested that the X94 line subsamples the NMC population (consistent with a previous report) since 30–40% of recorded L5 SST cells exhibited an NMC phenotype (*Figure 1—figure supplement 1c–g*). Aligning biocytin reconstructions of L5 MCs and NMCs revealed that these two populations have strikingly complementary vertical profiles of neurite density: MCs primarily innervated layers 1, 2/3 and 5, and NMCs primarily innervated L4 and the L5/L6 border (*Figure 1c*). Although some is known about their differing anatomical and physiological features (*Muñoz et al., 2017*; *Ma et al., 2006*; *Nigro et al., 2018*), relatively little is understood about how these two sub-classes of SST neurons, especially NMCs, might differentially integrate into and influence the cortical excitatory network.

To begin to answer this question, we first asked whether MCs and NMCs receive different patterns of excitatory inputs across the cortical layers. We transgenically expressed ChR2 in cortical excitatory neurons across all layers, and used scanning photostimulation to map the spatial profile of excitatory inputs to NMCs and MCs (*Figure 1d,e,f*; *Figure 1—figure supplement 4*). Remarkably, we found that L5 MCs and NMCs receive inputs from highly specific and largely non-overlapping sources. MCs, but not NMCs, frequently received excitatory input from either upper L5, L2/3, or from both L2/3 and L5, but received little input from L4 or L6, broadly consistent with prior studies (*Apicella et al., 2012*; *Jiang et al., 2015*; *Anastasiades et al., 2016*; *Kapfer et al., 2007*). In contrast, NMCs received strong input from L4 and/or the L5B/L6 border (*Figure 1e,f*; input from L4 and L6 was 62 ± 7% of total input for n = 14 NMCs versus 25 ± 3% for n = 15 MCs; p=6.5 · 10⁻¹⁰; two sample T-test; see also *Figure 1—figure supplement 3*) but relatively little input from L2/3 and L5. Thus, L5 MCs and NMCs appear to receive distinct and complementary patterns of excitatory innervation. An important caveat to these experiments is that our optogenetic stimulation did not recruit the same amount of activity in excitatory neurons across different layers (*Figure 1—figure supplement 4f*), presumably due to differences in intrinsic excitability or opsin expression levels. Although this makes it difficult to assess the relative strength of laminar input pathways in an absolute sense, stimulation of sites in non-preferred layers (e.g. stimulating L2/3 while recording from an L5 NMC) almost never evoked a response that was significantly greater than the background level of EPSCs, except at sites which were close to the borders between layers (*Figure 1—figure supplement 3k*). This suggests that input from most sites in non-preferred layers (L4 and L6 for MCs, L2/3 and L5A for NMCs) was either absent or too small to be detected via this method.

The striking laminar differences in inputs to NMCs and MCs suggested that they might be differentially recruited by activity of different cortical layers. To test this possibility, we specifically photostimulated L4 excitatory neurons via Cre-dependent expression of ChR2 in *Scnn1*-Cre mice (crossed to GIN or X94; *Figure 2a*). L4-specific photo-stimulation (with two different stimulus protocols, across four different intensities) drove large EPSCs in NMCs but evoked little to no input in MCs under identical conditions (*Figure 2b,c*; *Figure 2—figure supplement 1*). Current clamp recordings under the same conditions showed that L4 photo-stimulation reliably drove spiking in L5 NMCs, but not in L5 MCs (*Figure 2b,d*; *Figure 2—figure supplement 1*) despite the fact that MCs are intrinsically more excitable than NMCs (*Figure 1—figure supplement 1*). The lack of evoked responses in MCs was not due to differences in the degree of L4 activation (see *Figure 2—figure supplement 1* for controls). Thus, these results indicate a stark difference between L5 NMCs and MCs: L4 densely innervates and powerfully drives firing in L5 NMCs, but not L5 MCs.

## Common input mapping reveals subnetwork structure in L5 SST cell output

We next asked whether NMCs and MCs also exhibit layer-specificity in their inhibitory outputs. Since SST cells have been implicated in generating feedback inhibition (*Adesnik et al., 2012*; *Kapfer et al., 2007*; *Silberberg and Markram, 2007*), we hypothesized that MCs and NMCs might target their inhibitory outputs in order to reciprocally inhibit the same PC populations that excite them. For example, NMCs but not MCs would inhibit L4 PCs, whereas MCs but not NMCs would

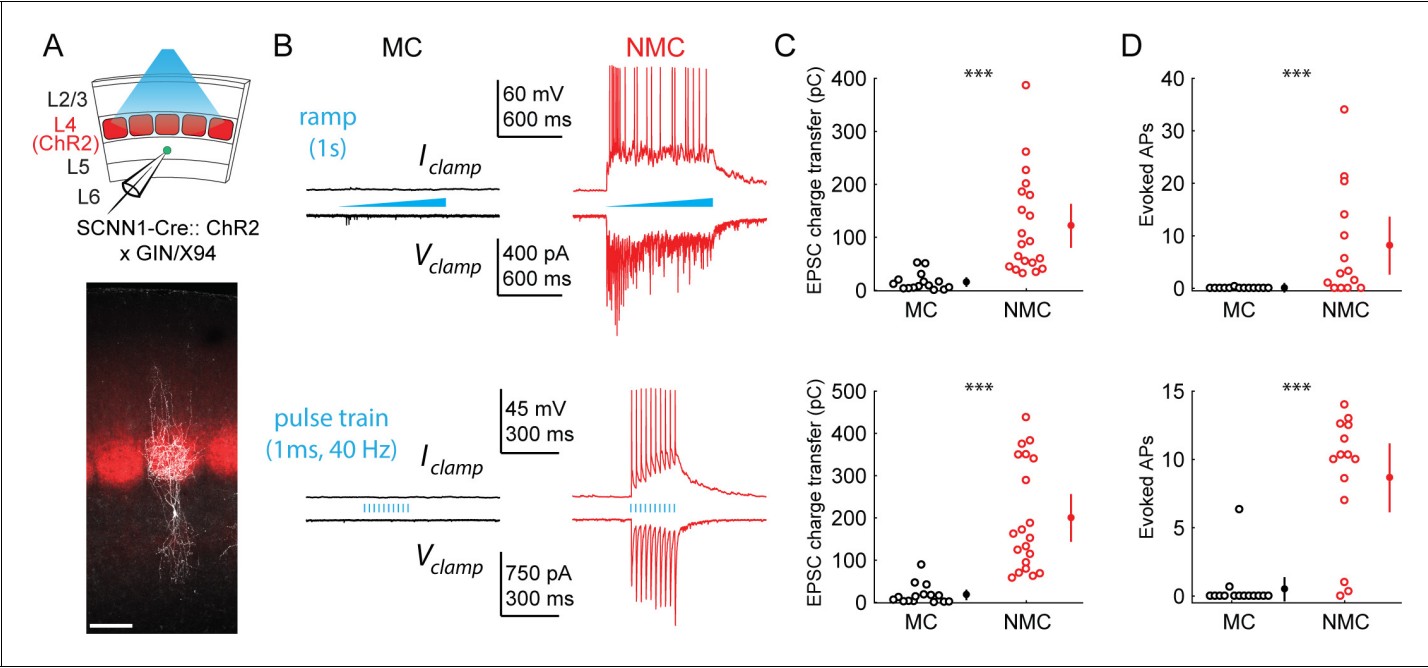

**Figure 2.** L4 photo-stimulation excites L5 NMCs but not L5 MCs. (A) Top: Schematic of the experimental configuration. L5 × 94 or GIN cells were recorded during photo-stimulation of L4 excitatory neurons. Bottom: Confocal image of a filled L5 × 94 neuron (white) with ChR2-TdTomato expression (red) visible in L4. Scale bar: 150 µm. (B) Top row: Example traces recorded in the current clamp (upper traces) or voltage clamp (lower traces) configurations during a 1 s ramp photo-stimulation. Bottom row: As above, but for photo-stimulation with a 40 Hz train of ten 1 ms pulses. (C) Quantification of excitatory charge transfer during maximum intensity 1 s ramp stimulation trials. Mean 122 ± 41 pC in n = 20 NMCs versus 15 ± 8 pC in n = 15 MCs; p=3.9 · $10^{-6}$, Wilcoxon rank sum test. (D) Quantification of the mean number of evoked action potentials during maximum intensity 1 s ramp stimulation trials. Mean 8.1 ± 5.5 spikes per trial in n = 15 NMCs versus 0.03 ± 0.05 spikes per trial in n = 15 MCs; p=6.6 · $10^{-4}$, Wilcoxon rank sum test. (E) As in C, for maximum intensity 40 Hz pulse train stimulation. Mean 200 ± 56 pA in n = 20 NMCs versus 18 ± 12 pA in n = 15 MCs; p=2.8 · $10^{-6}$, Wilcoxon rank sum test. (F) As in D, for maximum intensity 40 Hz pulse train stimulation. Mean 8.7 ± 2.4 spikes per trial in n = 15 NMCs versus 0.5 ± 0.9 spikes per trial in n = 15 MCs; p=1.5 · $10^{-6}$, Wilcoxon rank sum test. Error bars denote mean ±95% confidence interval. Three asterisks denotes p<0.001.

DOI: https://doi.org/10.7554/eLife.43696.007

The following figure supplement is available for figure 2:

**Figure supplement 1.** Responses of L4 and L5 SST cells to L4 photo-stimulation.
DOI: https://doi.org/10.7554/eLife.43696.008

inhibit L5 PCs. Alternatively, MCs, NMCs, or both cell types could globally target PCs within and across layers non-selectively. To address this, we used two photon optogenetic circuit mapping to determine whether the outputs of individual SST cells (in the non-specific SST-Cre line) diverge onto PCs in multiple layers. If individual SST cells target either L4 or L5 PCs, but not both, then we should never observe common input to pairs of L4 and L5 PCs when photo-stimulating single SST neurons. This can be tested by mapping optogenetically evoked unitary SST inhibitory connections onto multiple PCs recorded simultaneously and analyzing the spatiotemporal coincidence of evoked IPSCs onto different pairs of PCs, thereby measuring the amount of common input shared between pairs of PCs in different layers (*Yoshimura et al., 2005*; *Morgenstern et al., 2016*), (*Merel et al., 2016*). Although this approach does not discriminate between MCs and NMCs directly, it performs a more stringent test by extending our hypothesis to apply to the structure of the outputs of the L5 SST population as a whole, rather than the sparser subsets set labeled in the GFP lines. To maximize the spatial precision of photo-stimulation we used a soma-targeted opsin (*Figure 3a,b,c*) (*Baker et al., 2016*) and computer-generated holography (*Figure 3a,b*; *Figure 3—figure supplement 1a*). Since SST → PC synapses are often located on the distal dendrites of PCs, we recorded IPSCs using a cesium-based internal solution, and took additional steps to minimize false negatives (see Materials and methods). Using this method, we simultaneously mapped SST inputs to pairs of L4-L5 PCs and L5-L5 PCs (*Figure 3d,e,f*; *Figure 3—figure supplement 2f*).

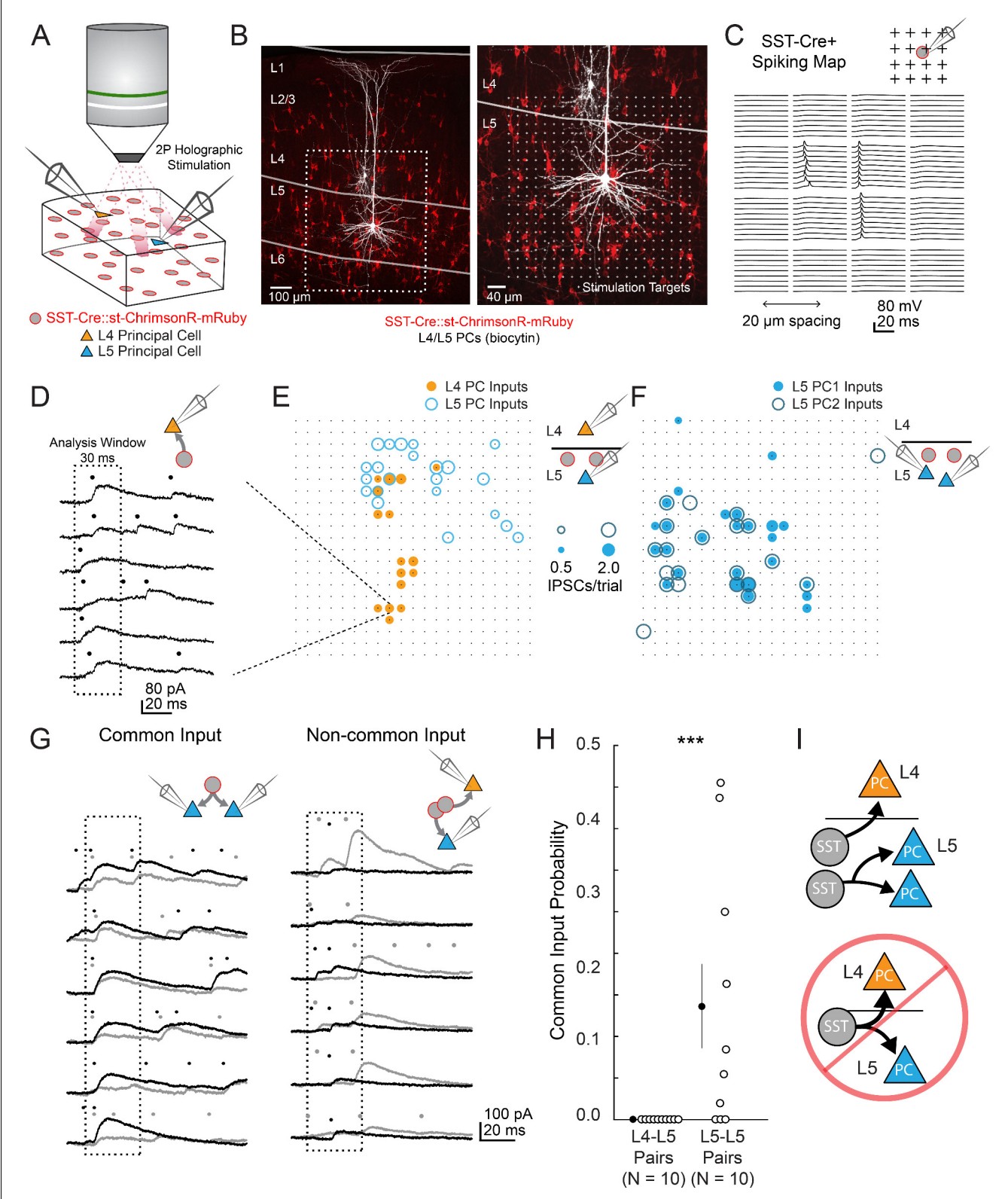

**Figure 3.** two photon optogenetic circuit mapping reveals that L4 and L5 PCs are inhibited by separate populations of L5 SST cells. (**A**) Schematic of the experimental configuration. IPSCs are recorded from a pair of PCs (either an L4/L5 pair or an L5/L5 pair) while SST cells expressing soma-targeted ChrimsonR-mRuby2 are focally activated using 2P photo-stimulation and computer generated holography. (**B**) Left: post-hoc confocal image showing SST cells expressing soma-targeted-ChrimsonR-mRuby2 (red) and biocytin fills of recorded PCs in L4 and L5 (white) at 10x magnification. Right:
*Figure 3 continued on next page*

*Figure 3 continued*

Confocal image at 20x magnification showing the grid of photo-stimulated target locations. Both images are max z-projections over 100 µm. (**C**) Spatial photo-excitation profile of a soma-targeted-ChrimsonR-mRuby2 expressing SST cell. Whole cell current-clamp recordings from this cell showing multiple trials of photo-stimulation at a 4 × 4 subsection of the photo-stimulation grid with 20 µm spacing between stimulation locations. The SST cell is recruited to spike only at a small number of stimulation sites, but does so reliably and with low jitter across trials at these sites. (**D**) Example traces showing IPSCs recorded from an L4 PC during SST photo-stimulation at a single site (corresponding to black boxed square in **E**) over multiple trials. Dots above each trace indicate the onset time of detected IPSCs (p=0.0003, Poisson detection). (**E**) Example overlay of maps showing the mean number of IPSCs at detected input locations during photo-stimulation for a simultaneously recorded L4 PC-L5 PC pair. Bubble size indicates the mean number of IPSCs evoked (deviation from background rate) per trial. (**F**) As in E, but for an L5 PC-L5 PC pair. (**G**) Example traces illustrating method for detection of common SST-mediated inputs to pairs of simultaneously recorded PCs. Left: IPSC traces at a single site recorded simultaneously in two PCs (each PC is indicated by black or grey traces) and corresponding detected IPSCs. IPSCs with synchronous onset occur in many trials, despite the trial-to-trial jitter in IPSC onset, suggesting that a SST cell which diverges onto both recorded PCs is being stimulated at this site (p=0.0005, synchrony jitter test). Right: IPSC traces from a different site. Evoked IPSCs are observed in both cells, but the lack of synchronicity suggests they arise from separate, neighboring SST cells (p=0.4). Dots above each trace indicate the estimated onset time of detected IPSCs. (**H**) Probability of detecting common SST input per photo-stimulated site for pairs consisting of L4 PCs and L5 PCs versus pairs consisting of two L5 PCs. No common input locations were detected in n = 10 L4-L5 pairs versus 13.7 ± 5.1% of all input locations locations stimulated in n = 10 L5-L5 pairs; p=1.1 · 10–3, Wilcoxon rank sum test Data are summarized by mean ±S.E.M. (**I**) Schematic of main result for SST outputting mapping. Individual L5 SST cells form inhibitory connections onto L4 PCs and or L5 PCs but not both.

DOI: https://doi.org/10.7554/eLife.43696.009

The following figure supplements are available for figure 3:

**Figure supplement 1.** Excitation profiles of st-ChrimsonR-expressing SST cells in multiphoton holographic SST-Cre mapping experiments.

DOI: https://doi.org/10.7554/eLife.43696.010

**Figure supplement 2.** Data processing and additional results for multiphoton SST output mapping.

DOI: https://doi.org/10.7554/eLife.43696.011

In L4-L5 PC pairs we observed very little common input when photo-stimulating SST neurons, but substantial common input in L5-L5 pairs (2.4 ± 1.3% spatially coincident inputs out of all input locations in n = 10 L4-L5 pairs, versus 28 ± 6.7% in n = 10 L5-L5 pairs; p=1.2 · $10^{-3}$, Wilcoxon rank sum test). Given that occasionally more than one SST cell might be photostimulated at any given target location (*Figure 2—figure supplement 1d,g*), we employed a statistical test for fine time scale synchrony of IPSCs between the patched cells at each candidate location (where both cells received input) to determine whether the IPSCs truly arose from a single SST cell diverging onto both recorded PCs (*Amarasingham et al., 2012*, *Figure 3g*, *Figure 3—figure supplement 2g,h,i,j*). Using this far more conservative test for the detection of common input, we detected no locations in which stimulation evoked common inputs for L4-L5 pairs, whereas we detected at least one common input in 7 of 10 L5-L5 pairs (*Figure 3h*; no locations in n = 10 L4-L5 pairs versus 13.7 ± 5.1% of all input locations in n = 10 L5-L5 pairs; p=1.1 · $10^{-3}$, Wilcoxon rank sum test; see also *Figure 3—figure supplement 2k*). These data argue that individual L5 SST cells connect to either L4 PCs or to L5 PCs, but never to both. In other words, L4 PCs and L5 PCs are inhibited by non-overlapping subnetworks of L5 SST cells.

## Paired recordings show dense, reciprocal, and selective intra- and translaminar connectivity

To unequivocally confirm the input/output mapping suggested by the optogenetic data presented above, we made paired intracellular recordings between both types of SST neurons and PCs in L4 and L5. We targeted L5 MCs and NMCs with the GIN and X94 lines as above, but also used the SST-TdT line to identify L5 SST cells more generally, and classified SST-TdT cells as putative MCs or NMCs based on their electrophysiological properties (*Figure 1—figure supplement 1g*; *Table 1*) and/or morphology. We observed extremely frequent L5 NMC → L4 PC connections (36/67 pairs tested; 54%; *Figure 4a,b*), even across long inter-somatic distances (183 ± 67 µm, mean ±S.D.; *Figure 4—figure supplement 1*), suggesting that L5 NMCs connect densely onto L4. In the opposite direction, we also frequently observed monosynaptic excitatory connections from L4 PCs onto L5 NMCs, consistent with the optogenetic experiments above (39/72 pairs tested; 54%). These synapses exhibited profound short-term facilitation during sustained high-frequency firing in the presynaptic cell (*Figure 4a*; *Figure 4—figure supplement 3*), which is a hallmark of excitatory connections onto SST cells, (*Kapfer et al., 2007*; *Beierlein et al., 2003*; *Berger et al., 2009*) including NMCs

**Table 1.** Connection rates for MCs and NMCs recorded in different transgenic lines; related to *Figure 4*.

Left columns show paired recording data collected using the GIN and X94 lines to respectively target MCs and NMCs in L5. Right columns show the same data and additionally include data collected using the SST-TdT line, with L5 SST cells classified as putative MCs or NMCs based on their intrinsic properties. Columns not displaying p values show the number and fraction of SST-PC pairs in which a monosynaptic connection was detected for a given condition.

| | GIN + X94 | | | GIN + X94+classified SST-TdT | | | All SST-TdT |
| --- | --- | --- | --- | --- | --- | --- | --- |
| | MCs | NMCs | P | MCs | NMCs | P | |
| L5SST→L4PC | 4/47 9% | 21/34 62% | $1.6 \cdot 10^{-4}$ | 4/68 6% | 36/67 54% | $2.0 \cdot 10^{-5}$ | 15/55 27% |
| L4PC→L5SST | 0/50 0% | 13/27 48% | $5.8 \cdot 10^{-4}$ | 1/95 1% | 39/72 54% | $<10^{-5}$ | 27/91 30% |
| L5SST→L5PC | 19/38 50% | 1/38 3% | $2.8 \cdot 10^{-4}$ | 24/46 52% | 2/65 3% | $<10^{-5}$ | 19/67 28% |
| L5PC→L5SST | 2/22 9% | 0/33 0% | 0.1431 | 4/29 14% | 1/60 2% | 0.02 | 5/55 9% |

DOI: https://doi.org/10.7554/eLife.43696.016

(*Beierlein et al., 2003*; *Hu and Agmon, 2016*; *Tan et al., 2008*). In cases where we tested connectivity bidirectionally, we frequently observed reciprocal connections (23/56 pairs tested; 41%). Thus, L5 NMCs and L4 PCs form a translaminar feedback inhibitory motif. We also observed frequent connections from L5 NMCs onto L4 fast-spiking (FS) cells (12/23 pairs tested; 52%; *Figure 4—figure supplement 2*), similar to a known circuit in which L4 non-Martinotti SST cells inhibit L4 FS cells (*Ma et al., 2006*; *Xu et al., 2013*).

In contrast, we almost never observed monosynaptic excitatory connections from L4 PCs to L5 MCs (1/95 pairs tested; 1%; *Figure 4a,b*) or from L5 MCs onto L4 PCs (4/68 pairs tested; 6%), despite the fact that these pairs were separated by smaller inter-somatic distances than L4 PC - L5 NMC pairs (143 ± 47 μm, mean ±S.D.; *Figure 4—figure supplement 1*). In a subset of these experiments, we recorded from L4 PCs in the voltage clamp configuration at +10 mV (using a cesium-based internal solution), but did not observe connections any more frequently (0/38 pairs tested; 0%). These data suggest that L5 NMCs are integrated into the densely interconnected network of L4 PCs and interneurons (*Beierlein et al., 2000*; *Petersen and Sakmann, 2000*), whereas L5 MCs are essentially isolated from it.

We next sought to confirm the notion raised by our 2P mapping experiments that L5 MCs would specifically and reciprocally connect to L5 PCs, while NMCs would not. Indeed, we observed frequent inhibitory connections from L5 MCs onto L5 PCs (24/46 pairs tested; 52%; *Figure 4c,d*), in agreement with prior literature (*Jiang et al., 2015*; *Fino and Yuste, 2011*; *Berger et al., 2010*). We also observed excitatory connections from L5 PCs onto L5 MCs, albeit more rarely (4/29 pairs tested; 14%; *Figure 4c,d*) but at a rate consistent with the literature (*Jiang et al., 2015*; *Levy and Reyes, 2012*). In contrast, we detected very few inhibitory outputs from L5 NMCs onto L5 PCs (2/65 pairs tested; 3%; *Figure 4c,d*) or excitatory connections from L5 PCs onto L5 NMCs (1/60 pairs tested; 2%; *Figure 3c,d*), despite the fact that L5 PCs were on average located much closer to L5 NMCs than were L4 PCs. The L5 PCs we recorded from in these experiments were sampled from throughout L5A and L5B, and included both thick and slender tufted PCs. The surprising dearth of intralaminar connectivity between L5 PCs and L5 NMCs stands in stark contrast to the dense intralaminar connectivity observed between L5 PCs and L5 MCs, as well as in other inhibitory circuits (*Fino and Yuste, 2011*; *Packer and Yuste, 2011*; *Levy and Reyes, 2012*). Furthermore, our finding that MCs and NMCs specifically target L5 PCs and L4 PCs very closely replicates results from a recent study (*Nigro et al., 2018*), lending further support to the notion that MCs and NMCs are wired into selective subnetworks with distinct laminar populations.

## CRE-DOG enables genetic access to subtypes of SST cells

The highly specific connectivity revealed by our circuit mapping experiments suggests that MCs and NMCs are specialized for different functions in cortical computation. If this is the case, we would expect that manipulating the activity of these groups of interneurons will have different effects on

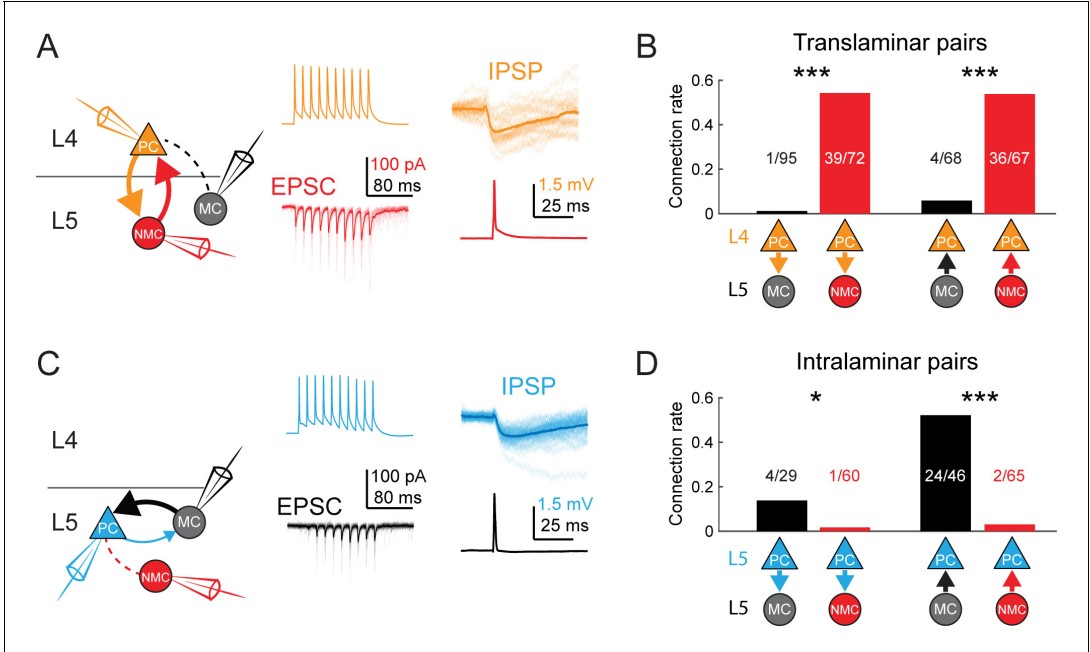

**Figure 4.** MCs and NMCs exhibit different patterns of monosynaptic connectivity with L4 and L5 PCs. (A) Paired recordings of L4 PCs (orange) and L5 NMCs/MCs (red/black).). Left: schematic of the tested circuit. Middle: example traces of evoked spikes in a L4 PC (orange) and the excitatory synaptic current in a L5 NMC (red). Right: example traces of evoked IPSPs in a L4 PC (orange) in response to a single action potential in a L5 NMC (red). (B) Bar graph summarizing translaminar connection rates between L4 PCs and L5 MCs (black bars) and L4 PCs and L5 NMCs (red bars). $p<10^{-6}$ for L4PC→L5MC (n = 95 connections tested onto 39 MCs) versus L4 PC→L5 NMC connection rate (n = 72 connections tested onto 51 NMCs); $p=2\cdot10^{-6}$ for L5MC→L4PC (n = 68 connections tested from 35 MCs) versus L5 NMC →L4PC connection rate (n = 67 connections tested from 51 NMCs); Monte Carlo permutation test. (C) As in A, but intralaminar pairs between L5 MCs/NMCs and L5 PCs (blue). (D) As in B, but for intralaminar connections with L5 PCs. $p=0.020$ for L5 PC→L5 MC (n = 29 connections tested onto 20 MCs) versus L5 PC→L5 NMC connection rate (n = 60 connections tested onto 35 NMCs); $p<10^{-6}$ for L5 MC→L5 PC (n = 46 connections tested from 30 MCs) versus L5 NMC →L5 PC connection rate (n = 65 connections tested from 37 NMCs); Monte Carlo permutation test. See also *Figure 4—figure supplement 1–3* and *Table 1*.

DOI: https://doi.org/10.7554/eLife.43696.012

The following figure supplements are available for figure 4:

**Figure supplement 1.** Distances of connections tested in paired recordings.

DOI: https://doi.org/10.7554/eLife.43696.013

**Figure supplement 2.** L5 NMC connectivity onto L4 FS cells.

DOI: https://doi.org/10.7554/eLife.43696.014

**Figure supplement 3.** Synaptic properties of L5 SST connections .

DOI: https://doi.org/10.7554/eLife.43696.015

cortical dynamics. Based on our circuit mapping results (*Figure 3*, *Figure 4*), we hypothesized that increasing NMC activity optogenetically would primarily affect L4, whereas increasing MC activity would impact neurons in L5, but not in L4. Since no recombinase driver line is available for NMCs, we sought to instead use the GFP lines themselves for selective expression of ChR2. To do this, we employed the CRE-Dependent-on-GFP (CRE-DOG) system which uses two split fragments of Cre recombinase, that unite as a functional Cre molecule only in the presence of GFP (*Tang et al., 2015*). We co-injected AAVs to drive expression of the CRE-DOG system, along with an AAV driving flexed ChR2-TdT, into X94 and GIN mice in order to target ChR2-TdT to GFP +SST cells in these mice.

In X94 mice injected with this cocktail (referred to hereafter as X94-ChR2 mice) we observed revealed a bright band of TdT +axonal arborization in L4, indicating effective labeling of GFP +NMCs (*Figure 5a,b*). While nearly all TdTomato expression colocalized with GFP, we observed a small number of GFP-/TdT +neurons, which mostly appeared to be pyramidal cells. This off-target expression is probably the result of CRE-DOG leakage, since injecting GFP- wildtype animals in the same manner also results in sparse expression of TdT in cortical neurons (*Figure 5—figure*

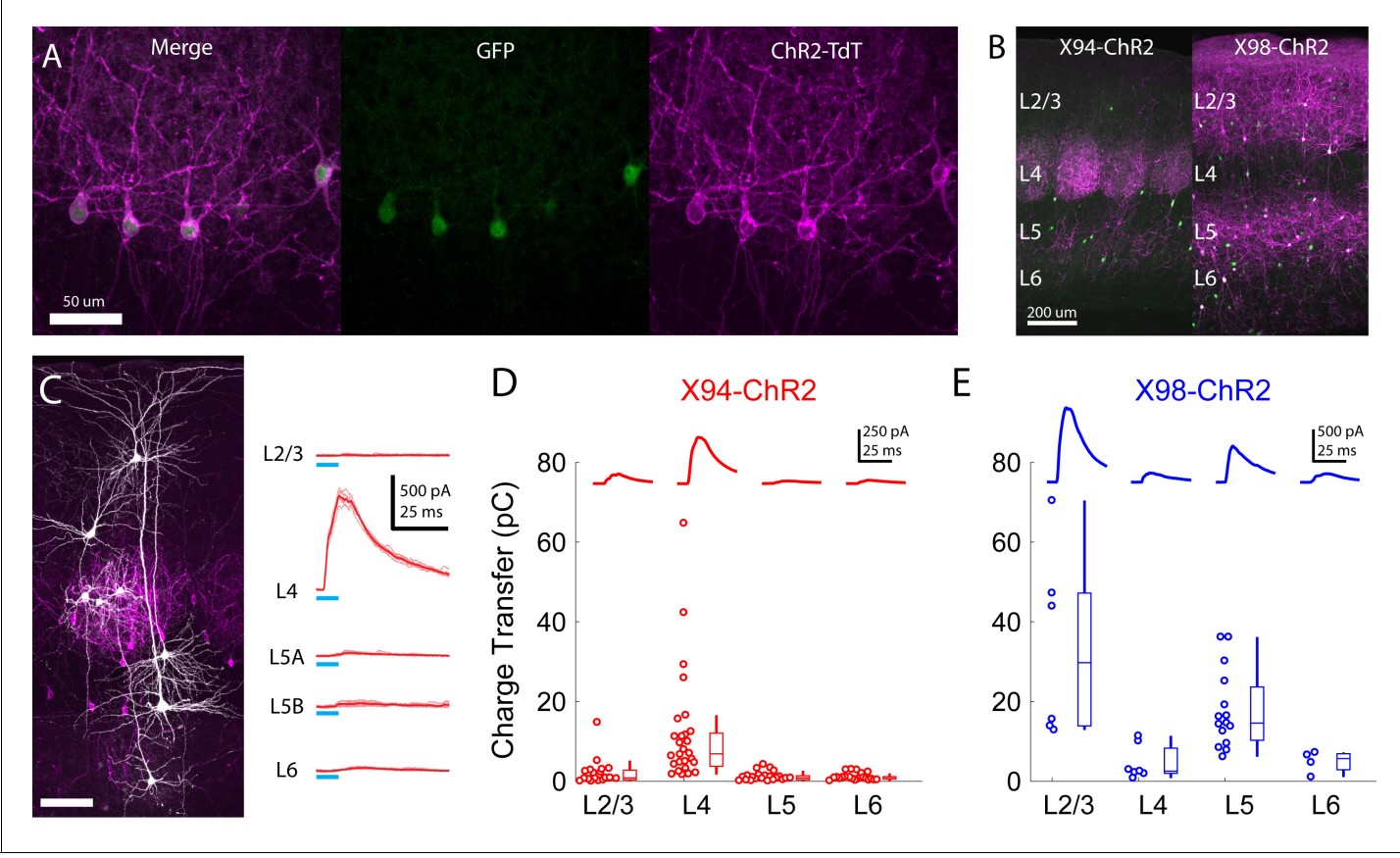

**Figure 5.** Cre-DOG enables optogenetic control of SST subtypes targeting different cortical layers. (**A**) Confocal image of cortical section from an X94 mouse injected with Cre-DOG AAVs (AAV2/8.EF1a.C-CreintG WPRE.hGH and AAV2/8. EF1a. N-Cretrcintc WPRE.hGH) along with AAV9.CAGGS.Flex. ChR2-tdTomato.WPRE.SV40. Left: X94- GFP cells (green). Middle: ChR2-TdT expression (magenta). Right: Merged image (**B**) Side by side comparison of X94-ChR2 mice and X98-ChR2 mice showing laminar differences in localization of ChR2-TdT + axons (**C**) Recording light-evoked IPSCs in X94-ChR2 slices. Left: post-hoc confocal image showing recorded neurons (white) and ChR2-TdT + NMCs (magenta). Right: example traces of light-evoked IPSCs recorded in neurons in different layers (**D**) Median charge transfer of evoked IPSCs in each PC recorded in X94-ChR2 slices, grouped by layer and accompanied by box and whisker plots. Top inset: grand average IPSC (**E**) As in D, but for X98-ChR2 mice.

DOI: https://doi.org/10.7554/eLife.43696.017

The following figure supplement is available for figure 5:

**Figure supplement 1.** Validation of Cre-DOG for optogenetic manipulation of SST subtypes in X94, GIN, and X98 mice.

DOI: https://doi.org/10.7554/eLife.43696.018

supplement 1c). However, most (232/267; 87%) TdT +neurons were GFP+, indicating that CRE-DOG allows expression of ChR2-TdT in GFP +X94 cells with high specificity. Furthermore, immunostaining confirmed that the large majority (240/267; 90%) of TdT +cells expressed SST (*Figure 5—figure supplement 1d,e*).

To confirm the efficacy and selectivity of optogenetically activating NMCs, we recorded IPSCs from PCs in layers 2–6 while photo-stimulating X94 cells with blue light (*Figure 5c,d*). Consistent with our previous experiments, NMC photostimulation reliably evoked powerful IPSCs in L4 PCs, whereas PCs in other layers usually received small IPSCs or showed no response (mean charge transfer ±C.I: L2/3 = 2.0 ± 1.4 pC; L4 = 10.7 ± 4.8 pC; L5 = 0.8 ± 0.5 pC; L6 = 1.1 ± 0.4 pC). At the population level, the evoked IPSC was only significant in L4 PCs (L2/3: p=0.41, n = 20; L4: $p<10^{-8}$, n = 30; L5: p=0.67, n = 27; L6: p=0.84, n = 22; F-test on linear mixed-effects model compared to baseline charge), though a small number of individual L2/3 PCs did exhibit substantial IPSCs. While this was expected for L2/3 and L5 PCs, the lack of evoked inhibition in L6 PCs is notable, since it suggests that the L6 to NMC connection is asymmetric, unlike the highly reciprocal connectivity

pattern seen between L4 PCs and NMCs. It is important to note that we illuminated all layers non-selectively in these experiments (e.g. photostimulated X94 cells both within and outside of L5, though these likely have similar connectivity; see Supplementary note, *Figure 1—figure supplement 3*) in order to mimic the pattern of activation we would expect to achieve during in vivo photostimulation in ensuing experiments. We conducted these experiments while pharmacologically blocking glutamatergic synaptic transmission, but observed very small or no EPSCs in response to photostimulation in a subset of experiments carried out without glutamatergic blockade (*Figure 5—figure supplement 1f*). These results indicate that CRE-DOG can be used with the X94-GFP line to achieve specific optogenetic control of NMCs.

Unfortunately, we found that GIN mice were unsuitable for specifically labeling MCs since we unexpectedly observed very bright TdT expression in a large number of L6 neurons (*Figure 5—figure supplement 1j*) that were electrophysiologically and morphologically different from MCs (*Figure 5—figure supplement 1l*) and negative for Cre expression in SST-Cre mice (*Figure 5—figure supplement 1m*). The GIN line also labels a heterogeneous population of SST cells (including some L4 NMCs; see *Table 2*). We therefore turned to an alternative GFP line, X98, which also labels MCs and not NMCs (*Ma et al., 2006*). An important caveat of this strategy is that the population of MCs labeled by the X98 line differs from that labeled by the GIN line (*Ma et al., 2006*), which makes it difficult to draw a direct link between the circuits described above and X98-based in vivo manipulation of MCs. Nevertheless, we reasoned that X98-ChR2 mice would provide a useful comparison to X94-ChR2 experiments, since it would allow us to target a distinct but similarly sized population of non-NMC SST cells using exactly the same protocol used to target X94 cells.

Injecting CRE-DOG/flexed-ChR2-TdT in X98 mice yielded expression which was strikingly complementary to the expression of X94-ChR2, with bright axonal fluorescence in L1, L2/3, and L5, but virtually none in L4 (*Figure 5b*). X98 has been described as labeling MCs primarily in deep L5 and upper L6 (*Ma et al., 2006*), as well as in L2/3. We observed a large number of TdT +neurons in L2/3 and throughout L5. In X98 mice injected with CRE-DOG/flexed-ChR2-TdT (X98-ChR2 mice), a substantial fraction of TdT +neurons (81/204; 40%) lacked visible GFP expression, but immunohistochemical staining for somatostatin showed that nearly all TdT +neurons (192/204; 94%), including GFP- neurons, were somatostatin positive; this discrepancy might arise from some SST cells expressing GFP only transiently during development. Slice recordings revealed that photostimulation of these neurons drove strong inhibition in L2/3 and L5 PCs, but relatively little in L4 and L6 PCs (mean charge transfer ±C.I: L2/3 = 33.9 ± 18.7 pC; L4 = 4.7 ± 3.0 pC; L5 = 17.9 ± 5.0 pC; L6 = 4.9 ± 2.7 pC; *Figure 5d*). In contrast to X94-ChR2, in X98-ChR2 slices evoked responses at the population level were only significant in L2/3 and L5 PCs (L2/3: $p<10^{-11}$, n = 6; L4: p=0.46, n = 7; L5: $p<10^{-6}$, n = 15; L6: p=0.53, n = 4; F-test on linear mixed-effects model). These results, along with the patterns of axonal fluorescence, suggests that ChR2-TdT + cells in X98-ChR2 mice are a population of SST cells which includes MCs but not NMCs, making X98-ChR2 mice a useful comparison for X94-ChR2 mice. As before, we illuminated the entire slice in these experiments, meaning that MCs in L2/3 were also photostimulated. Since L2/3 MCs inhibit L5 PCs and L5 MCs inhibit L2/3 PCs (*Jiang et al., 2015*; *Jiang et al., 2013*; *Lee et al., 2015*), the inhibition we observed in L2/3 and L5 PCs likely reflects contributions from both L2/3 MCs and L5 MCs.

**Table 2.** Summary of expression in SST reporter lines.

Four mouse reporter lines were used in this study to target SST neurons and subtypes. Each row provides a description of the expression observed in the barrel cortex in a particular layer for each reported line.

|  | SST-TdT | GIN | X94 | X98 |
|---|---|---|---|---|
| L2/3 | All SST cells | Dense, MCs | Very sparse | Sparse, MCs |
| L4 | All SST cells | Sparse, NMCs | Dense, NMCs | Very sparse |
| L5 | All SST cells | Moderate, preferentially in 5A, MCs | Moderate, preferentially in 5B, NMCs | Dense, preferentially in 5B, MCs |
| L6 | All SST cells | Sparse, Dim labeling of (non-SST?) cells | Sparse, preferentially in upper 6A, NMCs | Moderate |

DOI: https://doi.org/10.7554/eLife.43696.019

## SST subtypes drive layer-specific effects during active sensation

To test whether these two different SST subtypes differentially impact sensory processing across the cortical layers, we next optogenetically stimulated X94 and X98 cells while recording barrel cortex activity while animals actively touched a stimulus bar placed in different locations in their whisking field. A simple prediction based on our circuit mapping data is that these two subtypes of SST cells should suppress different cortical layers: NMCs should suppress L4, whereas MCs should suppress L2/3 and L5. However, since SST neurons can also disinhibit PCs by suppressing PV cells, it is possible that the net impact on PCs in different layers could instead be to increase activity. Furthermore, a previous in vitro study showed that PV cells are more effectively inhibited by NMCs than by MCs (*Xu et al., 2013*); thus a second hypothesis is that activating NMCs would cause a net disinhibition, whereas MCs would result in a net suppression.

Recording from X94-ChR2 mice, we observed that photostimulating NMCs powerfully suppressed the spontaneous activity of all units in L4, both FS (*Figure 6—figure supplement 2*; 5.2 ± 1.7 Hz control versus 1.1 ± 0.5 Hz light; p<10$^{-6}$, n = 39 L4 FS units), and RS units (1.8 ± 1.2 Hz control versus 0.9 ± 0.7 Hz light; p<0.001, n = 15 L4 RS units; *Figure 6a,b,e,f*; *Figure 6—figure supplement 1*). Similarly, stimulating NMCs strongly attenuated the response to sensory stimulation in L4 units (L4 RS: 3.8 ± 1.7 Hz control versus 2.3 ± 1.2 Hz light; p<10$^{-6}$, n = 15; L4 FS: 7.1 ± 4.0 Hz control versus 2.9 ± 1.0 Hz light; p<10$^{-6}$, n = 39; *Figure 6c,d,g,h*; *Figure 6—figure supplement 1*). This indicates that, in these conditions, enhancing NMC firing potently suppresses L4 and does not cause a net disinhibition of L4 excitatory neurons. NMC photostimulation caused little to no change in the activity of the L5 RS and L6 RS populations (L5 RS spontaneous: 5.1 ± 1.2 Hz control versus 5.1 ± 1.3 Hz light; p=0.20, n = 59; L6 RS spontaneous: 1.9 ± 1.0 Hz control versus 1.6 ± 0.7 Hz light; p=0.42, n = 13; L5 RS sensory-driven: 5.9 ± 1.2 Hz control versus 5.6 ± 1.3 Hz light; p<10$^{-6}$, n = 59; L6 RS sensory-driven: 2.1 ± 1.0 Hz control versus 1.9 ± 0.9 Hz light; p=0.04, n = 13), although some individual L5 RS units exhibited substantial increases or decreases in their firing rates. This is consistent with the lack of NMC inhibitory connections to PCs in these layers and supports the notion that NMC-mediated inhibition has layer-specific effects on cortical dynamics. Although our in vitro data did not reveal a strong monosynaptic connection from NMCs to L2/3 PCS, NMC photostimulation also robustly reduced spontaneous and sensory-evoked activity in the L2/3 RS population (L2/3 RS spontaneous: 0.9 ± 0.5 Hz control versus 0.1 ± 0.1 Hz light; p<10$^{-4}$, n = 10; L2/3 sensory-driven: 3.4 ± 1.4 Hz control versus 1.5 ± 0.5 Hz light; p<10$^{-6}$, n = 10), as well as that of nearly all FS units, including those outside of L4 (L2/3 FS sensory-driven: 8.3 ± 4.0 Hz control versus 5.6 ± 3.0 Hz light; p<10$^{-6}$, n = 11; L5 FS sensory-driven: 7.6 ± 2.8 Hz control versus 4.5 ± 3.0 Hz light; p<10$^{-6}$, n = 38; L6 FS sensory-driven: 8.3 ± 3.8 Hz control versus 7.0 ± 4.4 Hz light; p<10$^{-6}$, n = 9; *Figure 6—figure supplement 2*). Because L4 is an important source of excitatory drive to L2/3 PCs and FS cells (*Pluta et al., 2015*), the most likely explanation is that that NMC photoactivation indirectly reduces activity in L2/3 by dramatically reducing excitatory input from L4.

Photostimulating SST cells in X98-ChR2 mice yielded dramatically different effects. The activity of the L2/3 and L5 RS populations was substantially reduced both during spontaneous conditions (L2/3 RS spontaneous: 0.8 ± 0.6 Hz control versus 0.3 ± 0.2 Hz light; p<10$^{-6}$, n = 29; L5 RS spontaneous: 5.4 ± 1.5 Hz control versus 4.0 ± 1.8 Hz light; p<10$^{-6}$, n = 42; *Figure 6a,b,e,f*; *Figure 6—figure supplement 1*), and during sensory stimulation (L2/3 RS sensory-driven: 2.4 ± 0.8 Hz control versus 1.2 ± 0.4 Hz light; p<10$^{-6}$, n = 29; L5 RS sensory-driven: 6.4 ± 1.6 Hz control versus 4.5 ± 1.8 Hz light; p<10$^{-6}$, n = 42; *Figure 6c,d,g,h*; *Figure 6—figure supplement 1*), whereas the activity of the L4 RS population showed no change or small reductions under the same conditions (L4 RS spontaneous: 1.1 ± 0.4 Hz control versus 1.2 ± 0.5 Hz light; p=0.82, n = 12; L4 RS sensory-driven: 2.3 ± 0.8 Hz control versus 2.0 ± 0.6 Hz light; p<0.01, n = 12). We also observed a substantial increase in the firing of L5 RS units following photostimulation (*Figure 6—figure supplement 1*); interestingly, we noted that this rebound effect was also present in L5 RS units recorded in X94-ChR2 mice, though we did not analyze it further here. X98-ChR2 photostimulation did not cause a significant effect in the L6 RS population (L6 RS spontaneous: 4.3 ± 4.7 Hz control versus 3.2 ± 2.5 Hz light; p=0.13, n = 6; L6 RS sensory-driven: 4.6 ± 5.0 Hz control versus 4.1 ± 4.1 Hz light; p=0.24, n = 6), though we sampled few L6 units. As with X94-ChR2 mice, we also observed a global suppression of FS units across all layers when photostimulating in X98-ChR2 mice (*Figure 6—figure supplement 1*); however, the magnitude of FS suppression was somewhat smaller in X98-ChR2 mice relative to X94-

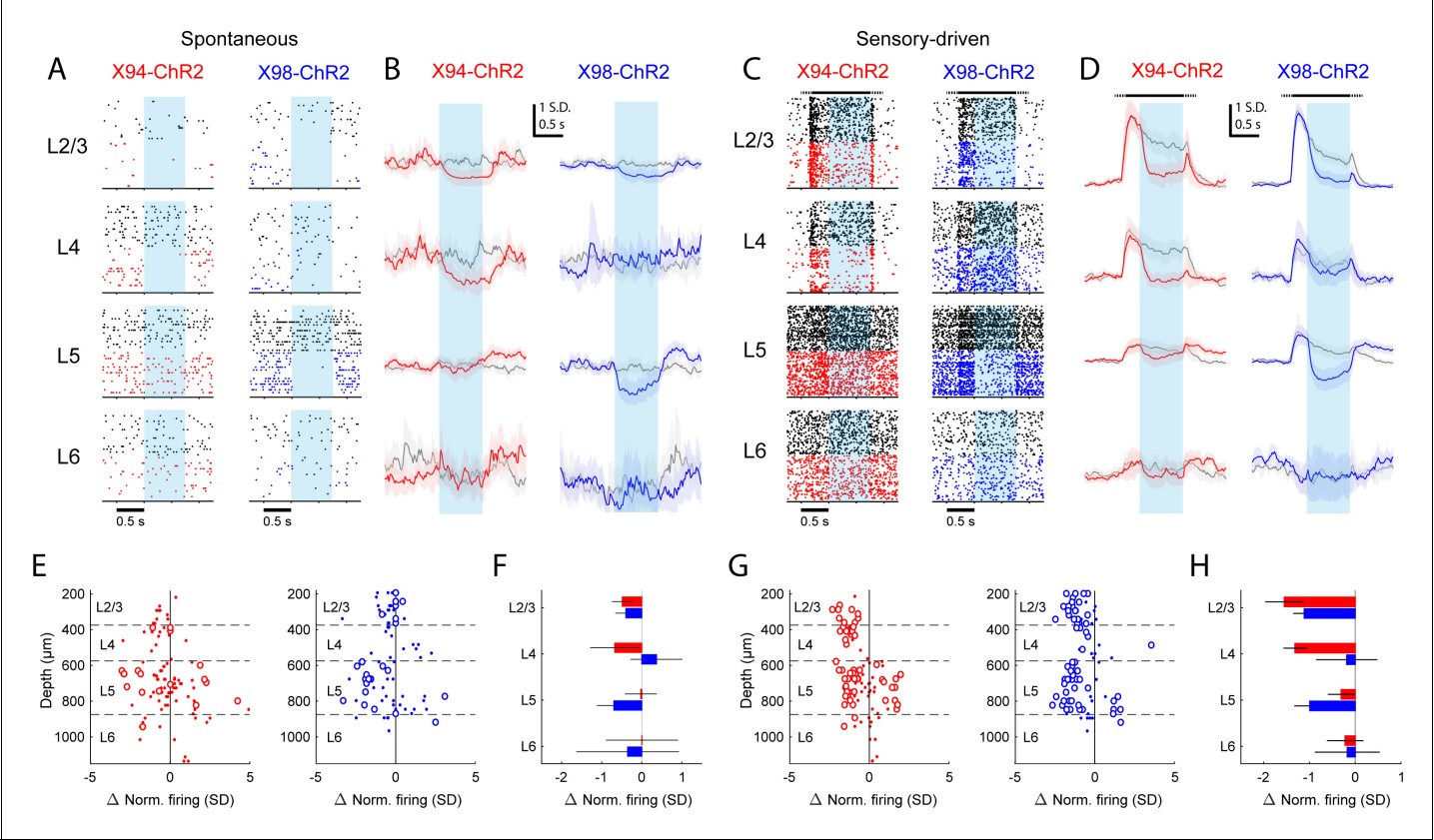

**Figure 6.** Differential layer-specific modulation of cortical activity by optogenetic activation of NMCs and MCs in vivo. (**A**) Raster plots showing activity in example RS units recorded from different layers in X94-ChR2 and X98-ChR2 mice. Black rasters show trials with no stimulus that is spontaneous activity. Colored rasters show trials with photostimulation of X94-ChR2 (red) or X98-ChR2 (blue). Light blue region indicates photostimulation period. (**B**) Grand averages of z-scored RS unit activity in L2/3, L4, L5, and L6 showing spontaneous activity (gray) and activity on photostimulation trials (red, X94-ChR2; blue, X98-ChR2). Responses have been smoothed with a 100 ms alpha kernel and downsampled to 50 Hz. Shaded regions indicate 95% confidence interval. (**C**) As in A, but for sensory-driven activity from trials in which a vertical pole is presented to the whiskers as a tactile stimulus (**D**) As in B, but for sensory-driven activity (**E**) Change in normalized spontaneous firing of RS units versus depth below pia for X94-ChR2 (left, red) and X98-ChR2 (right, blue) mice. Large circles and small dots indicate units that were respectively significantly or not significantly modulated by optogenetic stimulation. (**F**) Mean change in normalized firing rate by layer for X94-ChR2 (red bars) and X98-ChR2 (blue bars). Errorbars indicate 95% confidence interval. (**G**) As in E, but for change in sensory-driven activity (**H**) As in F, but for change in sensory-driven activity.

DOI: https://doi.org/10.7554/eLife.43696.020

The following figure supplements are available for figure 6:

**Figure supplement 1.** PSTHs of RS unit sensory responses during in vivo optogenetic manipulation of MCs and NMCs.
DOI: https://doi.org/10.7554/eLife.43696.021

**Figure supplement 2.** PSTHs of FS unit sensory responses during in vivo optogenetic manipulation of MCs and NMCs.
DOI: https://doi.org/10.7554/eLife.43696.022

ChR2 mice (L2/3 FS sensory-driven: 5.2 ± 4.2 Hz control versus 2.4 ± 1.6 Hz light; $p<10^{-6}$, n = 10; L4 FS sensory-driven: 7.2 ± 1.9 Hz control versus 5.4 ± 1.6 Hz light; $p<10^{-6}$, n = 47; L5 FS sensory-driven: 7.1 ± 1.9 Hz control versus 5.5 ± 1.9 Hz light; $p<10^{-6}$, n = 34; L6 FS sensory-driven: 7.8 ± 2.8 Hz control versus 5.6 ± 2.1 Hz light; $p<10^{-6}$, n = 6).

Taken together, these data suggest that the activation of SST neurons in X98-ChR2 mice exerts a wholly different effect on the cortical microcircuit than that of SST neurons in X94 mice, which labels NMCs. Furthermore, we did not observe any effect on cortical activity when we repeated these experiments in wild-type mice (injected with the same viral cocktail used with X94/X98 mice), indicating that the effects we observed depended specifically on the optogenetic stimulation of GFP +SST cells (*Figure 6—figure supplement 1g–j*). Two important caveats should be noted in interpreting these results in the context of our L5 MCs circuit mapping data: 1) photostimulation in these

experiments activates X98 cells both in L5 and L2/3, and 2) the cells labeled in X98-ChR2 represent a population that is different from the population labeled by the GIN line, which might be expected to exhibit different connectivity (*Morishima et al., 2017*). However, taken together with our X94-ChR2 data, these results demonstrate that SST subtypes are specialized to modulate specific cortical layers and suggest that MCs and NMCs exert very different effects on cortical activity.

## Single-cell RNA sequencing maps NMCs onto transcriptomic clusters

Prior studies using transcriptomic approaches have identified multiple clusters within the cortical SST population (*Tasic et al., 2016*; *Tasic et al., 2018*). However, very few molecularly identified SST cell types have been mapped onto physiological/functional phenotypes in the brain (*Paul et al., 2017*). Basing on their striking physiological and circuit differences, we next asked whether NMCs might be transcriptionally distinct. To address this we performed single-cell RNA sequencing on tdTomato[+] or GFP[+]/ tdTomato[+] cells isolated by fluorescence activated cell sorting from S1 cortex of SST-Cre; LSL-tdTomato; X94-GFP triple transgenic mice. Clustering tdTomato[+] and GFP[+]/ tdTomato[+] cells together based on the 1000 top variable genes yielded 10 distinguishable clusters of SST[+] neurons (*Figure 7a*). These clusters showed remarkable correspondence to clusters similarly generated from single-cell RNA-seq on SST[+] neurons from primary visual cortex (V1sp) and anterior lateral motor cortex (ALM) (*Figure 7—figure supplement 1*), supporting the idea that SST cell types are conserved across cortical regions (*Tasic et al., 2018*). We then asked whether single-cell RNA-seq could

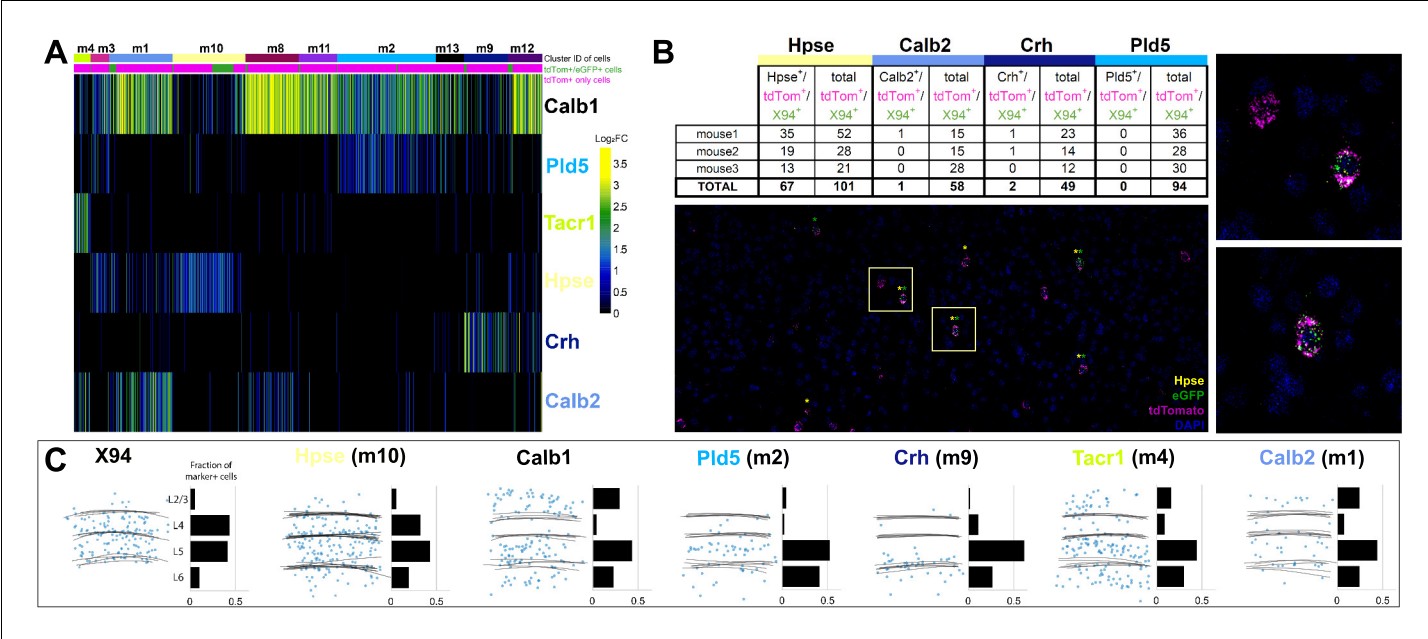

**Figure 7.** Single cell RNA sequencing of X94 and SST cells. (A) Single-cell RNA-seq was performed on SST-TdTomato[+] and GFP[+]/tdTomato[+] cells FACS-purified from primary somatosensory cortex of *X94-eGFP;Sst-cre;LSLtdTomato* mice. Cells were clustered using the Louvain algorithm and organized into vertical columns based on their cluster identity (top bar), with distribution of GFP+/tdTomato +cells indicated below. Horizontal rows correspond to mRNA expression for highly differentially expressed genes that were selected as cluster classifiers. (B) Triple-label RNA in situ hybridizations were performed on *X94-eGFP;Sst-cre;LSLtdTomato* mice to validate the predictions made by single-cell RNA-seq. The table shows quantitation of cells co-labeled with probes for selected marker genes, GFP and tdTomato (a proxy for *Sst* expression). Representative image shows overlapping signals from cluster classifier *Hpse*, GFP and tdTomato. Insets show examples of triple-positive cells at higher magnification. (C) Summary of laminar distribution of X94 cells cluster classifier/tdTomato double-positive cells based on tracing and scoring positions of labeled cells across three animals for each condition. Horizontal lines represent estimated positions of laminar boundaries. Left-hand panel shows localization of X94 NMCs using anti-GFP for X94 cells and anti-dsRed for Sst-tdTomato cells. Histograms give normalized frequency values for cluster classifier+/tdT +cells for each indicated cortical layer.

DOI: https://doi.org/10.7554/eLife.43696.023

The following figure supplement is available for figure 7:

**Figure supplement 1.** Comparison of cortical SST neuron clusters predicted by two independent analyses.
DOI: https://doi.org/10.7554/eLife.43696.024

distinguish X94-GFP$^+$ cells. Indeed,~84% (157 out of 188) GFP$^+$/SST$^+$ neurons were distributed among three clusters, with just over half of all GFP cells (102 cells) falling within a single cluster (cluster m10) that could be defined by the specific expression of the gene *Hpse* (*Figure 7A*). Although cluster m10 contains a small number of cells expressing *Calb1*, they are distinct from X94-GFP$^+$ cells and *Hpse*-expressing cells, consistent with the finding that X94 cells do not co-localize with calbindin immunostaining (*Figure 7a*) (*Ma et al., 2006*). Significant numbers of GFP$^+$ cells were also found in clusters m1 (32 cells or 17%) and m12 (23 cells or 12%), however, suggesting that the population of cells labeled in the X94 transgenic mouse line may in fact be somewhat heterogeneous.

We next performed triple-label RNA in situ hybridizations using cluster-specific marker genes to validate and characterize the SST cell clusters predicted by single-cell RNA-seq. Tissue sections were prepared from brains of SST-Cre; LSL-tdTomato; X94-GFP triple transgenic mice and hybridized to probes for genes differentially-expressed in selected SST clusters together with probes for GFP and tdTomato to identify SST$^+$ cells. The numbers of marker gene/GFP/tdTomato triple-positive and GFP/tdTomato double-positive cells were scored from tissue sections from S1 cortex of three mice. The results of this analysis together with a representative image showing Hpse/GFP/tdTomato triple-label RNA in situ hybridization are shown in *Figure 7b*. The co-localization of *Hpse* expression in ~67% of GFP/tdTomato-positive cells validates the assignment of ~half of X94-GFP cells to cluster m10 based on single-cell RNA-seq. Similarly, few if any GFP-expressing cells co-express *Crh* or *Pld5*, markers for clusters in which X94-GFP cells are largely absent. The inclusion of 16% of GFP$^+$/SST$^+$ cells in cluster m1 is curious, given the expression of *Calb2* – a MC marker – by most cells in this cluster. However, single-cell RNA-seq indicates that X94-GFP cells do not express *Calb2*, a conclusion supported by the paucity of Calb2/GFP/tdTomato triple-positive cells by RNA in situ hybridization (1/58 cells; *Figure 7b*).

We next analyzed the localization of cells expressing markers for 5 of the 10 major SST cell clusters predicted from single-cell RNA-seq to determine whether molecularly-defined SST neurons correspond to cells with distinct laminar positions in S1 cortex (*Figure 7c*). Notably, *Hpse* – a marker for the cluster m10, the main X94-GFP-containing cluster – labels SST-cre;tdTomato$^+$ cells found primarily within L4 and L5, similar to the laminar distribution of X94 cells. Crh$^+$/tdTomato$^+$ cells (cluster m9) were found mostly in deep L5/upper L6 and Pld5$^+$/tdTomato$^+$ cells (cluster m2) in mid-L5; Tacr1$^+$/tdTomato$^+$ cells (corresponding to cluster m4) were distributed broadly across all laminae. Calb2$^+$ cells (cluster m1) colocalizing with tdTomato were found to be broadly distributed among all layers except L4, which instead is largely occupied by *Hpse* neurons. Taken together, these data strongly suggest that *Hpse* defines the L4/L5 NMC cells, and further support the idea that the transcriptomically defined SST neurons described here represent biologically meaningful sub-classes with distinct characteristics based on their anatomy, morphology, connectivity and physiology.

## Discussion

Despite recent strides in understanding cortical inhibitory circuitry, many key features remain unknown. Our data establish the existence of two subnetworks of SST interneurons that make exquisitely selective and reciprocal interactions with different sets of cortical layers. Optogenetic circuit mapping shows that L5 MCs receive excitatory inputs chiefly from PCs in L2/3 and L5, the primary cortical output layers, while L5 NMCs receive inputs mainly from PCs in L4 and upper L6, the primary input zones for afferent input from the ventral posteromedial thalamus (*Wimmer et al., 2010*). Paired recordings and 2-photon holographic optogenetic interrogation indicate that, in turn, these same SST subtypes selectively inhibit the same PC populations that excite them, at least within L4 and L5. In vivo, NMCs and MCs differentially suppress the activity of specific cortical layers. Thus NMCs and MCs are functionally segregated into two distinct networks with selective and complementary laminar connectivity, and functional impacts in the awake brain. Transcriptome profiling further suggests that SST neurons break down into as many as 10 sub-clusters that might compose unique neocortical inhibitory microcircuits. More specifically, our data point to a transcriptionally distinct subset of SST neurons (referred to here at 'NMCs' but marked by the selective expression of the gene *Hpse*) that powerfully controls PC activity in Layer 4, potentially gating bottom up input into the cortex. Conversely, several transcriptionally distinct subsets of SST neurons, commonly referred to as 'MCs', have no direct impact on L4, and instead potently control supragranular and infragranular PCs, the major targets of top down input from other cortical areas.

These results reveal a previously unknown, striking degree of specificity in the inhibitory cortical wiring diagram. In particular, the observation that L5 NMCs exhibit nearly no intralaminar connectivity with L5 PCs, but do engage in dense, reciprocal connectivity with L4 PCs is inconsistent with the idea of a single, global blanket of SST-mediated inhibition. Instead, SST-PC circuits appear to more closely resemble a patchwork quilt, comprised of multiple networks of SST subtypes which independently modulate separate spatial domains. SST-PC connectivity can be extremely dense and non-selective within one of these domains (e.g. creating a blanket within a single laminar microcircuit), but highly selective on the scale of layers and columns.

## Functional implications of separate, layer-specific SST feedback circuits

The striking difference in the input and output circuitry of SST subtypes suggested that these two interneuron classes might have different functional effects during sensory processing. Indeed, our Cre-DOG optogenetics experiments demonstrate that MCs nearly exclusively suppress the cortical output layers that they innervate, namely, L2/3 and L5. Conversely, NMCs potently suppress L4, but have a minimal impact on L5 activity. Notably, NMC photo-stimulation did cause a prominent deactivation of L2/3 cells. Since our slice data indicate that NMCs make only weak or infrequent connections onto L2/3 PCs, much of this reduction in L2/3 activity can likely be attributed to a loss of input from L4, which we previously showed is necessary for their sensory response. Nevertheless, a component of this reduction could be due to direct monosynaptic inhibition from NMCs (*Figure 5d*).

In our previous work, we showed that direct optogenetic suppression of L4 PCs resulted in deactivation of FS units, and net disinhibition and increased sensory evoked activity of L5 RS units. Based on this work we proposed that L4 PCs exert an inhibitory influence on L5 PCs via a disynaptic circuit in which L4 PCs drive L5 FS cells, which in turn inhibit L5 PCs. Consistent with this, here we saw that FS units in L5 showed strong and consistent deactivation when L4 was suppressed by NMC photostimulation. However, despite this reduction in L5 FS unit activity, we did not observe a net disinhibition across the population of recorded L5 RS units – in fact, somewhat remarkably, despite the major changes in both L4 and L2/3, on average we observed no net change in L5 RS unit activity across the recorded population. We can consider several possible explanations for this seeming discrepancy. Photostimulating NMCs, as we did here, resulted in a much stronger suppression of L4 than we achieved in our previous work with direct optogenetic suppression of L4 PCs. This may be due to the partial efficacy of the silencer eNpHR3.0, together with the fact the the Cre line we use in our previous study for L4 PCs (scnn1ta-tg3-Cre) only labels a fraction of the L4 PC population. Owing to the near complete suppression of L4 PC activity in this study, we also observed a much large deactivation of L2/3 RS units than in our prior study, where L2/3 was only mildly, albeit significantly, impacted. Thus, one possibility is that themuch stronger suppression of L2/3, which should cause a profound reduction in L2/3 to L5 excitatory drive that counteracted the loss of L4-driven disynaptic inhibition. Alternatively, photoactivation of NMCs might cause presynaptic inhibition of glutamatergic release at intracortical/thalamocortical synapses onto L5 PCs (*Urban-Ciecko et al., 2015*) or even drive the release of neuropeptides that suppress L5 cells through pre- or post-synaptic mechanisms. Further work is needed to reconcile these results. Our circuit mapping experiments revealed both MCs and NMCs exhibit dense, reciprocal connectivity with specific excitatory populations. Thus, both SST subtypes participate in potent recurrent inhibitory loops that might be critical for network stabilization, gain control, or competitive interactions between neural ensembles within each layer. Interestingly, NMCs and MCs also receive different sources of long-range input: NMCs but not MCs receive direct thalamic input from VPM, whereas MCs are known to receive long-range inputs from primary motor cortex (*Tan et al., 2008*; *Hu and Agmon, 2016*; *Ji et al., 2016*; *Kinnischtzke et al., 2014*; *Cruikshank et al., 2010*). Taken together, these circuit features suggest that NMCs and MCs might be specialized to regulate different streams of input to the cortical microcircuit. The major 'bottom-up' pathway to the barrel cortex is the lemniscal thalamocortical projection from VPM which carries spatiotemporally precise exteroceptive signals from the whiskers and which acts as a key driver of cortical activity via its projection to L4 and the L5B/L6 border. By virtue of their dense reciprocal connectivity with L4, as well as their direct connections from VPM (as well as cells at the L5B/L6 border, which are also VPM recipients), NMCs will generate feedback and feedforward inhibition targeted to L4 PCs. This inhibition is likely to occur on slower timescales than PV-mediated inhibition; thus, rather than enforcing temporally precise responses, NMC-mediated inhibition might control amplification of bottom-up sensory signals that is thought to occur in L4. In

contrast, MCs will have little effect on the integration of bottom-up signals in L4. However, other important long-range afferents to the barrel cortex, such as those from motor cortices, contralateral S1, and the paralemniscal pathway projection from POm, target the infragranular and supragranular layers (including L1). Inputs from these projections are thought to exert a primarily modulatory effect on barrel cortex activity, and in many cases might carry 'top-down' signals conveying contextual/predictive information (*Larkum, 2013*). MCs are positioned to directly influence the integration of these signals, and indeed MC-mediated inhibition is capable of gating dendritic integration of these inputs. The parallel structure of MC and NMC networks could allow S1 to independently gate its sensitivity to top-down inputs without changing its sensitivity to bottom-up inputs, and vice versa. In turn, differential modulation of these SST sub-circuits, either by local or long range excitatory input, by VIP-interneuron mediated inhibition (*Muñoz et al., 2017*; *Lee et al., 2013*; *Pfeffer et al., 2013*; *Pi et al., 2013*) or neuromodulation (*Muñoz et al., 2017*; *Xu et al., 2013*; *Polack and Contreras, 2012*; *Xiang et al., 1998*) could represent a mechanism by which the brain dynamically fine-tunes the balance between bottom-up and top-down information during sensory integration.

The stark differences in the inputs to MCs and NMCs suggest that they will likely exhibit different patterns of activity in vivo. For technical reasons, most studies of SST activity in awake animals have focused on SST cells in superficial cortical layers, which are mostly MCs. These studies have revealed that MCs have unique functional properties. Compared to other cell-types, L2/3 MCs in the barrel cortex are poorly driven or even hyperpolarized by single whisker sensory stimuli (*Gentet et al., 2012*), and appear relatively decoupled from spontaneous fluctuations shared by other neurons in L2/3 (*Sachidhanandam et al., 2016*). Additionally, along with some L5 MCs, L2/3 MCs are suppressed when the cortex enters an active state of arousal, whereas NMCs in L4 and L5 increase their activity (*Muñoz et al., 2017*; *Pala and Petersen, 2018*). These findings are consistent with other studies which have observed that subtypes of SST cells with either wide or narrow spike waveforms (which might correspond to MCs and NMCs) are differentially modulated by changes in arousal, behavior, and rewarding stimuli (*Kim, 2016*; *Kvitsiani et al., 2013*; *Reimer et al., 2014*). Changes in the activity of SST cells have been proposed to perform operations such as adjusting the gain of sensory responses or modulating dendritic integration (*Lee et al., 2012*; *Phillips and Hasenstaub, 2016*; *Murayama et al., 2009*); SST sub-circuits, like the ones described here, would allow for these operations to be applied to specific layers and/or cell-types. A pressing question for future investigation will be to determine how local and long-range connectivity contributes to the unique activity patterns of SST subtypes, and conversely, to determine how distinct SST subtypes differentially shape the dynamics of the cortical microcircuit in different sensory and behavioral contexts.

## Diversity of SST cells

Our physiological data support the notion of at least two major SST subclasses, defined by their input/output connectivity with L4 or L2/3 and L5. Our single-cell RNA-seq data identified 10 distinct SST subtypes that show good correspondence to 16 SST clusters obtained from single-cell RNA-seq analysis on SST neurons from V1 and ALM cortex (*Tasic et al., 2018*). Several of the clusters identified by our analysis split into multiple, smaller clusters in the analysis by *Tasic et al. (2018).*, likely due to greater resolution afforded by the greater sequencing depth and/or different parameters used for clustering in this latter study. A unique aspect of our approach is that we additionally included the X94-GFP transgene in our SST cell purification and sequencing, so that we could correlate the transcriptomic data with our physiological analysis of the X94 line. Together with triple-label RNA in situ hybridizations, our sequencing results strongly suggest that L4/L5 NMCs labeled in the X94 mouse line correspond to a transcriptomically distinct class of SST neurons characterized by *Hpse* expression. Our RNA in situ hybridization studies further demonstrate that transcriptomically defined SST subtypes show distinct cortical lamination patterns.

Previous reports have associated the anatomical location, axonal targeting patterns and physiological properties of a subset of Martinotti cells with expression of calretinin (encoded by the *calbindin2* or *Calb2* gene) (*Nigro et al., 2018*; *Paul et al., 2017*; *Hilscher et al., 2016*; *Xu et al., 2006*). Curiously, from single-cell RNA-seq studies, cluster SST-Etv1_3 in *Tasic et al. (2018).* - which corresponds to cluster m1 identified in this study - has been interpreted to represent this subset of Martinotti cells based largely on *Calb2* expression and enrichment in L2/3 and L5. This poses a conundrum, as we find that 16% of X94-GFP NMC cells cluster with putative *Calb2+* Martinotti cells in cluster m1/SST-Etv1_3 (*Figure 7a* and *Figure 7—figure supplement 1*). It should be noted,

however, that triple-label RNA in situ hybridizations confirm that the GFP$^+$ cells are depleted in *Calb2* expression (*Figure 7b*), suggesting that this cluster is heterogenous, at least with respect to *Calb2* expression. Moreover, *Calb2* expression is enriched in multiple clusters including not just m1 but also m4 and m8 (*Figure 7a*), precluding the reliance on this gene as a unique molecular marker of cells that are transcriptomically distinct. Future studies will be required to understand the apparent heterogeneity of what have been classically defined as Martinotti cells and whether their molecular identity and distinction from NMCs can be refined by a more granular analysis based on genome-wide transcriptomics.

More generally, an outstanding question for future work is to address how sub-classes of SST neurons identified by single cell transcriptomics differ from one another functionally, potentially due to their local and long-range connectivity and their sensitivity to different neuromodulators. Previous studies on MC-subtypes have outlined at least two distinct sub-classes of MCs (*Muñoz et al., 2017*; *Nigro et al., 2018*; *Morishima et al., 2017*), yet the functional roles of these putatively distinct MC subtypes remains essentially unknown. One route to address the functional implications of this high diversity of SST neurons is to use marker genes to generate intersectional driver lines (*Nigro et al., 2018*; *Paul et al., 2017*; *He et al., 2016*) that target each of the 10 clusters, which would greatly facilitate further physiological and anatomical analysis.

We find that NMCs comprise a large fraction of L5 SST cells and perhaps of SST cells more generally (see *Figure 1—figure supplement 1g*). If L5 NMCs are so prevalent, one might ask why they have been reported on only sparingly. Beyond studies which used the X94 line (*Ma et al., 2006*; *Tan et al., 2008*; *Hu and Agmon, 2016*), close examination of the literature reveals several reports of cells with L4-targeting morphologies and other properties characteristic of NMCs (*Fairén et al., 1986*; *Helmstaedter et al., 2009*; *Kumar and Ohana, 2008*; *Porter et al., 2001*), which have sometimes been called 'Lorente de Nó cells' in attribution to their earliest describer (*Cobas et al., 1987*; *Fairén, 2007*; *Lorente de Nó, 1992*). The previous lack of genetic tools to target these cells may have hindered detailed investigation of L5 NMCs until now. It is likely that NMCs in all layers have sometimes been misclassified as fast-spiking cells due to their quasi-fast-spiking intrinsic properties, especially in studies done without genetic or immunohistochemical markers for somatostatin and parvalbumin. This issue likely also applies to studies performing in vivo extracellular recordings, since spike waveforms of NMCs are similar to those of parvalbumin-expressing FS cells (*Kim, 2016*; *Kvitsiani et al., 2013*; *Muñoz et al., 2014*).

L5 NMCs bear striking resemblances to the SST cells in L4 which are also non-Martinotti cells and have sometimes been called 'low-threshold spiking' or 'LTS' cells (*Ma et al., 2006*; *Beierlein et al., 2003*; *Xu et al., 2013*; *Beierlein et al., 2000*; *Ma et al., 2012*; *Gibson et al., 2005*). Compared to L5 NMCs, L4 NMCs have similar morphologies and intrinsic properties (*Figure 1—figure supplement 3*), excitatory inputs (*Figure 1—figure supplement 3* and *Figure 2—figure supplement 1*), inhibitory targets (*Xu et al., 2013*) (*Figure 4—figure supplement 2*), and transcriptional profiles (*Figure 7*). Furthermore, we observed that excitatory synapses onto L5 NMCs exhibited what appeared to be asynchronous EPSCs during sustained high frequency firing in the presynaptic cell (*Figure 4—figure supplement 3c*). Asynchronous release of glutamate is uncommon in cortical circuits; to the best of our knowledge, the only other published observation is at the synapse from L4 PCs onto L4 SST cells (*Beierlein et al., 2003*).. Taken together, these similarities suggest that L4 and L5 NMCs are a single cell-type. Due to their lack of intralaminar connectivity, L5 NMCs might be thought of as 'ectopic' L4 NMCs; however, the number of NMCs in L5 is likely comparable to or perhaps even greater than the number of L4 NMCs in the barrel cortex. The X94 line labels ~ 15% of L5 SST cells, but this is likely a lower bound since this line does not provide complete coverage over L5 NMCs. Biocytin fills of L5 SST-TdT cells revealed that 19/52 cells (37%) possessed NMC morphologies, compared to 31/52 (60%) that possessed MC morphologies and 2/52 that could not be placed in either category. Consistent with this, a support vector machine trained to distinguish L5 GIN cells from L5 × 94 cells based on their intrinsic properties (*Figure 1—figure supplement 1g*) labeled 48% of L5 SST-TdT cells as putative NMCs. However, L5 NMCs often appeared to have larger cell bodies than MCs, which could potentially have made L5 NMCs easier to locate or patch and thereby introduced a bias in this assessment. Nevertheless, we estimate that 15–40% of L5 SST cells are NMCs. L4 and L5 respectively host approximately 10% and 40% of all SST cells in S1, so assuming nearly all L4 SST cells are NMCs (*Xu et al., 2013*), we estimate that L4 and L5 NMCs collectively represent roughly 15–25% of all SST cells in S1.

In this study we relied on four transgenic lines for SST neurons (SST-IRES-Cre, X94, X98 and GIN). Although the latter three GFP lines proved useful, heterogeneity within the GFP⁺ population within these lines (particularly the GIN line, see *Figure 1—figure supplement 3* and *Figure 2—figure supplement 1*) (*Halabisky et al., 2006*; *McGarry et al., 2010*) and our single-cell RNA-seq data imply that they each are likely to label a mixture of what may be functionally distinct SST subtypes, while other subtypes of SST cells are not covered by these lines at all, such as long-range projecting SST cells that are most prevalent in L6 (*He et al., 2016*). Importantly, our 2P mapping experiments (using the SST-Cre line) show that cells sampled from the SST population in an unbiased manner connect to L4 or L5 in a mutually exclusive manner; this argues that the distinction between NMCs and MCs is a genuine dichotomy which generalizes to the broader SST population. However, it is certain that these two groups, particularly MCs, can be further subdivided, and examining the connectivity of these finer SST subdivisions will very likely add further nuance to the scheme we describe here. For example, recent work suggests that specific subtypes of MCs receive different amounts of VIP innervation (*Muñoz et al., 2017*). Although we did not observe any obvious specificity in how MC circuits are organized with respect to L5A versus L5B and/or slender-tufted, intratelencephalic PCs versus thick-tufted, pyramidal tract PCs (data not shown), recent evidence suggests that some MC subtypes preferentially connect to L5 PC subtypes (*Morishima et al., 2017*; *Hilscher et al., 2016*). Achieving a full understanding of connectivity and functional interactions among SST and PC subtypes will require a more sophisticated understanding of the taxonomy of cortical cell-types, as well as the development of new genetic tools and circuit mapping methods.

Taken together, the data in this study establish two new fundamental inhibitory motifs in the cortex: two subnetworks of SST cells that interconnect with specific cortical compartments – the input and output cortical layers, potentially providing a means to fine tune cortical computation in the barrel cortex during different sensory or behavioral demands. Since most cortical regions appear to contain numerous subtypes of SST cells, we hypothesize that similar architectures will be present in other cortical regions, such as primary visual cortex. Consistent with this notion, other studies have shown that subtypes of SST cells with distinct morphologies, molecular and electrophysiological properties, and connectivity exist in the hippocampus (*Yuan et al., 2017*; *Harris et al., 2018*; *Müller and Remy, 2014*; *Lovett-Barron et al., 2012*). More generally, the approached we employed here to connect the anatomy, physiology, synaptic connectivity and transcriptional profile of specific neuronal subtypes may represent a generalizable strategy to define neuronal subtypes and reveal their unique contributions to brain activity and behavior. With respect to SST neurons' role in sensory computation, our data raise the possibility that sub-networks of dendrite-targeted interneurons fine tune the balance between bottom up and top down input in cortical processing.

## Materials and methods

All experiments were performed in accordance with the guidelines and regulations of the Animal Care and Use Committee of the University of California, Berkeley under protocol AUP-2014-10-6832-1.

**Key resources table**

| Reagent type (species) or resource | Designation | Source or reference | Identifiers | Additional information |
|---|---|---|---|---|
| Genetic reagent (Mus musculus) | Scnn1-tg3-Cre line | Jackson Labs | #009613 | |
| Genetic reagent (Mus musculus) | Emx1-IRES-Cre line | Jackson Labs | #005628 | |
| Genetic reagent (Mus musculus) | PV-IRES-cre line | Jackson Labs | #008069 | |
| Genetic reagent (Mus musculus) | SST-IRES-cre line | Jackson Labs | #013044 | |
| Genetic reagent (Mus musculus) | GIN line | Jackson Labs | #003718 | |

*Continued on next page*

*Continued*

| Reagent type (species) or resource | Designation | Source or reference | Identifiers | Additional information |
|---|---|---|---|---|
| Genetic reagent (Mus musculus) | X94-GFP line | Jackson Labs | #006334 | |
| Genetic reagent (Mus musculus) | X98-GFP | Jackson Labs | #006340 | |
| Genetic reagent (Mus musculus) | Ai9 Rosa-LSL-tdTomato line | Jackson Labs | #007909 | |
| Recombinant DNA reagent | AAV9.CAGGS.Flex. ChR2-tdTomato. WPRE.SV40 | University of Pennsylvania Vector Core | | |
| Recombinant DNA reagent | AAV9-2YF-hSyn-DIO-ChrimsonR-mRuby2-Kv2.1 | This lab | | Available at Addgene (Plasmid #105448); Described in *Pégard et al. (2017)* |
| Recombinant DNA reagent | AAV2/8.EF1a.C-CreintG.WPRE.hGH | Massachusetts Ear and Eye Infirmary Vector Core | | |
| Recombinant DNA reagent | AAV2/8.EF1a.N-Cretrcintc.WPRE.hGH | Massachusetts Ear and Eye Infirmary Vector Core | | |
| Antibody | Rat monoclonal anti-somatostatin primary | Millipore | MAB354 | 1:1000 dilution |
| Antibody | Goat polyclonal anti-rat Alexa 647 secondary | Life Technologies Corporation | A21247 | 1:200 dilution |
| Sequence-based reagent | tdTomato ISH probe | ACDBiotechne | 317041-C1 and C2 | |
| Sequence-based reagent | Calb2 ISH probe | ACDBiotechne | 313641 C1 | |
| Sequence-based reagent | Hpse ISH probe | ACDBiotechne | 412251-C1 | |
| Sequence-based reagent | Tacr1 ISH probe | ACDBiotechne | 428781 C2 | |
| Sequence-based reagent | Timp3 ISH probe | ACDBiotechne | 471311-C2 | |
| Sequence-based reagent | Pld5 ISH probe | ACDBiotechne | custom C2 | |
| Sequence-based reagent | Crh ISH probe | ACDBiotechne | 316091 C1 | |
| Sequence-based reagent | Calb1 ISH probe | ACDBiotechne | 428431 C2 | |
| Sequence-based reagent | eGFP-o4 ISH probe | ACDBiotechne | 538851-C3 | |

## Transgenic mice

The following mouse lines were used for this study: the Scnn1-tg3-Cre line (JAX stock # 009613), the Emx1-IRES-Cre line (JAX stock #005628), the PV-IRES-cre line (B6;129P2-Pvalbtm1(cre)Arbr/J; JAX stock #008069), the SST-IRES-cre line (JAX stock 013044), the GIN line (FVB-Tg(GadGFP)45704Swn/J; JAX stock #003718), the X94-GFP line (Tg(Gad1-EGFP)94Agmo/J; JAX stock 006334), the X98-GFP line (Tg(Gad1/EGFP)98Agmo/J); JAX stock 006340), the Ai9 Rosa-LSL-tdTomato line (JAX stock # 007909). Mice were housed in cohorts of five or fewer with a light:dark cycle of 12:12 hr, and were used for experimentation during their subjective night.

## Viral infection

Neonatal mice (p0-3) were deeply cryo-anesthetized and placed in a head mold. Viral aliquots were loaded into a Drummond Nanoject injector and injected into four sites in the barrel cortex of the left hemisphere. At each site, virus was injected at multiple depths (two depths for scnn1-tg3-cre, three depths for emx1-IRES-Cre and SST-IRES-Cre mice and for CRE-DOG injections) in increments of 18.4 nL or 36.8 nL (for SST-IRES-Cre and CRE-DOG injections), for a total of ~150–440 nL of virus injected per mouse. Following injections, mice were moved to an incubation chamber for recovery, and were returned to the dam once they regained color and began to move. Viruses used were AAV9. CAGGS.Flex.ChR2-tdTomato.WPRE.SV40 (acquired from the University of Pennsylvania Vector Core; undiluted for scnn1-tg3-cre, diluted 1:1 with PBS for emx1-IRES-Cre mice), AAV9-2YF-hSyn-DIO-ChrimsonR-mRuby2-Kv2.1, AAV2/8.EF1a.C-CreintG.WPRE.hGH and AAV2/8.EF1a.N-Cretrcintc. WPRE.hGH (acquired from the Massachusetts Ear and Eye Institute). For CRE-DOG experiments, we injected a mixture of 1 part AAV2/8.EF1a.C-CreintG.WPRE.hGH, one part and AAV2/8.EF1a.N-Cretrcintc.WPRE.hGH, and one part AAV9.CAGGS.Flex.ChR2-tdTomato.WPRE.SV40. In some initial slice experiments, we used a similar cocktail but with two parts AAV9.CAGGS.Flex.ChR2-tdTomato. WPRE.SV40.

## Brain slice recording

Acute thalamocortical slices were prepared from mice (ages p14-29, at least 14 days after viral injection) as previously described (*Adesnik and Scanziani, 2010*). Slices were placed in a recording chamber and constantly perfused with oxygenated artificial cerebro-spinal fluid (NaCl 119 mM, KCl 2.5 mM, $MgSO_4$1.3 mM, $NaH_2PO_4$1.3 mM, glucose 20 mM, $NaHCO_3$26 mM, $CaCl_2$2.5 mM) maintained at 32° C (21° C for multiphoton mapping experiments). Slices were oriented with the caudal surface facing up in the recording chamber. To ensure minimal disruption of vertical connectivity, all slices used for recording were inspected under infrared illumination at 40x magnification and/or post-hoc confocal imaging to confirm that pyramidal cell apical dendrites stayed roughly parallel with the surface of the slice or receded slightly deeper as they progressed apically. Whole cell recordings were performed using glass micropipettes (2–5 MΩ resistance) pulled on a Sutter P-1000 Micropipette Puller. Pipettes were filled with a $Cs^+$ based internal ($CsMeSO_4$135 mM, NaCl 8 mM, HEPES 10 mM, $Na_3GTP$ 0.3 mM, MgATP 4 mM, EGTA 0.3 mM, QX-314-Cl 5 mM, TEA-Cl 5 mM) or a potassium gluconate based internal (K-gluconate 135 mM, NaCl 8 mM, HEPES 10 mM, $Na_3GTP$ 0.3 mM, MgATP 4 mM, EGTA 0.3 mM). In some experiments, biocytin (0.4–1%) was dissolved into the internal solution to enable morphological recovery. Voltage recordings were not corrected for the junction potential. Series resistance was monitored with negative voltage steps during each trial, and was compensated up to 60%. Data were analyzed from recordings in which series resistance remained stable and below 30 MΩ. Data were acquired and filtered at 2.2 kHz using a Multiclamp 700B Amplifier (Axon Instruments) and digitized at 20 kHz (National Instruments). All data were acquired using custom written MATLAB (Mathworks) software.

## Characterization of intrinsic properties

In all recordings using K-based internal solution, an F-I curve was measured at the start of the experiment using a series of 1 s current injections, at −200 pA, −100 pA, and then proceeding in 50 pA increments from +50 to+500 pA. In some experiments, additional current steps were manually designated and performed online to aid in estimation of rheobase. Resting membrane potential was defined as the median membrane potential during a baseline period measured immediately after break-in. Input resistance was calculated with Ohm's law using the steady state membrane potential during subthreshold current injections steps (current clamp) and/or the steady state current during 5 mV voltage steps (voltage clamp). Action potential onset was detected using code adapted from the Berg lab's *Spike_threshold*_PS function, which defines onset as the point of maximum positive slope in the phase space of the membrane potential and its first derivative (*Sekerli et al., 2004*). Spike width was measured as the full-width of each spike at the voltage halfway between the action potential threshold and the peak amplitude (half-max). Rheobase was estimated using the average of 1) a linear fit (with coefficients constrained to be nonnegative using the *lsqnonneg* function in MATLAB) of the F-I relation during the last subthreshold current injection step and the first few suprathreshold steps and 2) linear extrapolation of the current necessary to reach threshold based

on measurements of the resting membrane potential, input resistance, and average threshold value of the first action potentials evoked during suprathreshold injections. These two measures were usually in good agreement. Adaptation index was calculated (following the Allen Brain Institute's Cell Types Database protocol) for each current injection using the expression:

$$\frac{1}{N-1}\sum_{n=1}^{N-1}\frac{ISI_{n+1} - ISI_n}{ISI_{n+1} + ISI_n}$$

Where N is the number of spikes during that current step and ISI is the interspike interval.

## Paired recording connectivity testing

We first targeted whole-cell recordings to a fluorescent (GFP +or TdTomato+) SST cell, and then subsequently patched nearby neurons in the same slice. In some cases, we recorded serially from several neurons while maintaining the recording of the first neuron, in order to test multiple connections. Monosynaptic excitatory connectivity onto SST cells was tested by driving trains of 10 spikes in the presynaptic cell at 70 Hz via current injection, while monitoring for EPSCs in the postsynaptic cell. Stimulation was repeated at least 15 times in all pairs tested. Monosynaptic inhibitory connectivity from SST cells onto other neurons was tested by driving spikes in the presynaptic cell while monitoring postsynaptically for IPSCs (Cs-based internal, postsynaptic cell held at +10 mV) or IPSPs (K-based internal, postsynaptic cell depolarized to approximately −52 mV). Electrical connectivity between SST cells was tested by hyperpolarizing each cell with 1 s current injections (at least 15 trials) while monitoring for hyperpolarization in the other cell.

For L5SST-L5PC pairs, we recorded from both pyramidal tract and intratelencephalic type PCs, which could be distinguished by their laminar positions (preferentially L5B versus L5A), morphology visualized via infrared (large soma versus smaller soma) and post-hoc confocal imaging (thick-tufted apical dendrites versus slender-tufted), and/or their intrinsic properties (initial burst/doublet spiking followed by non-adapting spikes versus continuously adapting regular-spiking phenotype (*Hattox and Nelson, 2007*; *Kim et al., 2015*; *Schubert et al., 2001*). We did not observe any significant differences in the connectivity of either L5PC type with L5MCs or NMCs. For L5SST-L4PC pairs, we did not distinguish between spiny stellate and pyramidal/star-pyramidal excitatory cells.

For paired recordings between L5 SST cells and L4 FS/PV cells, we identified FS/PV cells using PV-Cre; LSL-TdTomato mice in some experiments (*Figure 4—figure supplement 2i*). However, it was often difficult to visualize X94 cells using these animals due to the TdTomato fluorescence being much brighter than the GFP fluorescence. In other experiments (*Figure 4—figure supplement 2h*), we targeted FS/PV cells in L4 by looking for L4 neurons with large cell bodies under IR, and then confirmed the identity of these cells electrophysiologically, with the primary criteria separating them from being narrow spike widths (slightly shorter than the average NMC spike) and little or no spike frequency accommodation during high amplitude steps of current injection.

To classify SST-TdT cells as putative NMCs or MCs, we fit a support vector machine (cross validated 10-fold) to perform binary classification of L5 GIN cells and L5 × 94 cells using only their intrinsic electrophysiological properties. We found that a classifier based on only two measures (spike width and estimated rheobase) performed just as well as multivariate classification based on a large number of metrics (~85% accuracy).We then used this classifier to predict the identity of a different dataset of L5 SST cells recorded in SST-TdT mice. This approach is likely to have resulted in a small number of SST-TdT cells being misclassified; however, the connectivity of putative NMCs and MCs were highly similar to the connectivity of NMCs and MCs identified using the X94 and GIN lines. Furthermore, our conclusions about the differences in connectivity rates of L5 MCs and NMCs with L4 and L5 PCs are unchanged by the exclusion of the SST-TdT dataset, with the exception of L5PC→L5SST connections – a circuit which has been studied in some detail by others. This approach also effectively assumes a dichotomy in L5 SST cells, since we have only two labels (MC and NMC) to provide as training data, which is an important caveat since it is likely that further subdivisions of SST cells exist in L5 (*Ma et al., 2006*). In a handful of cases, we recorded from SST-TdT cells which appeared to be FS cells (*Hu et al., 2013*), with very narrow spikes, low input resistances, and a near complete lack of spike-frequency accommodation during high amplitude current injection steps; these neurons were excluded from further analysis.

Using paired recordings, we tested 544 total possible connections between 146 L5 SST cells (39 L5 GIN cells, 53 L5 × 94 cells, 54 L5 SST-TdT cells) and PCs/FS cells in L4 and L5. Data from a subset of these neurons (n = 17 L5 GIN cells) were included in a previous study (*Pluta et al., 2015*). This dataset was unbalanced, and because in some cases we tested multiple connections onto the same L5 SST cell, included some non-independent observations. Because of this, we used Monte Carlo permutation tests to test for significant differences between the connectivity rates of MCs and NMCs. We generated a permuted dataset with the same observation structure (same number of L5 SST cells and same number of connections tested per L5 SST cell) in place for MCs and NMCs by randomly resampling with replacement at both levels. We then measured the difference in observed connectivity rate for the MC and NMC groups, and repeated this procedure 100,000 times to generate a null distribution of rate differences. We used this distribution to perform a 1-tailed test for significant differences between MC and NMC connectivity rates for each type of connection tested (*Supplementary file 1*).

## Optogenetic connectivity mapping in vitro

Experiments were done in slices from Emx1-Cre; GIN or Emx1-Cre; X94 mice injected with an AAV driving Cre-dependent expression of ChR2 in all excitatory cells. Whole cell voltage clamp recordings were performed in GFP +L5 cells to target L5 MCs (Emx1-Cre; GIN) or L5 NMCs (Emx1-Cre; X94). A digital micromirror device was used to focally photo-stimulate excitatory cells in different regions of the slice in order to map the spatial profile of excitatory inputs to recorded MCs and NMCs.

Prior to experiments, slices were briefly visually inspected with epifluorescence under a 5x objective to confirm that a wide area containing dense, even expression of fluorescence (tagged to an opsin) was present in the barrel cortex. Recordings were targeted to within this region, which typically covered the entire lateral extent of barrel cortex in 4–5 slices. Slices in which expression appeared faint or uneven were discarded.

In some experiments, it was necessary to locate fluorophore-positive cells in slices also containing an excitatory opsin. To avoid excitotoxicity that can result from excessive illumination of opsin-containing neurons, we limited illumination to very brief intervals (1–2 s) while searching for fluorophore-positive cells. In some cases where the target cells were weakly fluorescent (young GIN and X94 animals), we searched for these cells while keeping the slice submerged in sucrose-substituted ACSF. Once target cells were located, this solution was washed out and replaced with normal recording ACSF prior to patching these cells and starting experiments.

## DMD-based excitatory input mapping

Laser light was generated using a 1W 445 nm diode laser (Ultralasers) and routed via a liquid light guide into a CEL5500 digital micromirror device (DMD) (Digital Light Innovations). The projection from the DMD was then collimated and integrated into the light path of the microscope, before being focused onto the slice chamber using a 5x (Olympus). For experiments using widefield illumination, the DMD passively reflected but not spatially modulate light. Prior to photo-stimulation, infrared and epifluorescence images were captured using an IR-1000 CCD camera (DAGE-MTI) and imported into MATLAB.

Excitatory mapping experiments were performed using a modified version of a previously described protocol (*Pluta et al., 2015*). Mapping was performed over an area extending from pia to the white matter, covering 2–4 barrel columns laterally (~400 to~800 μm). For mapping excitatory inputs to GIN and X94 cells, the DMD was used to pattern light into a square region (75 μm x 75 μm). Each stimulation site was spaced 40 μm apart from adjacent ones, resulting in some overlap of adjacent stimuli. We chose to 'ramp' our photostimulation, starting each stimulus with the light off and linearly increasing the light intensity over time. Ramping in this manner minimizes activation of fibers of passage (*Adesnik and Scanziani, 2010*). In each trial, a 'sawtooth' light stimulus composed of three successive 25 ms ramps of light (1.25 mW/mm$^2$ final intensity) was applied to one stimulus site (unlike in *Pluta et al. (2015)*, which used only a single ramp per trial). This protocol was chosen in order to maximize the short-term facilitation of excitatory inputs to L5 SST cells, though in practice we found it was usually possible to observe responses during the first ramp alone. Ten regions were stimulated per second in a serial, pseudorandom order, with 4 s breaks after every 10 s of mapping.

Control experiments were performed using identical stimulation conditions while recording from ChR2[+] neurons in all layers. These experiments determined the spatial resolution of photostimulation and confirmed that spiking was elicited in ChR2 +neurons only when regions very close to the soma were stimulated. We also included n = 2 experiments mapping inputs to L5 × 94 cells which were performed using the exact mapping protocol described in *Pluta et al. (2015)*, though our results and conclusions were not substantially altered by their exclusion.

All data were analyzed using custom written MATLAB software. Data preprocessing consisted of removing baseline offsets and slow fluctuations from recordings by subtracting a down-sampled and median-filtered versions. Charge was calculated as the integral of the preprocessed recordings during photo-stimulation and the subsequent 25 milliseconds. To aggregate maps across cells, we first rotated the average map collected in each experiment in order to horizontally orient the laminar boundaries of the mapped area. Maps were next translated vertically to align the L4-L5 laminar boundary, and translated horizontally to align either the home column or the soma position of the recorded cell, before being horizontally cropped to an area ±300 μm of their center and then averaged to yield a summary map. For the average map shown in *Figure 1—figure supplement 3k*, we first binarized each input map by performing a permutation test comparing the excitatory charge evoked at each stimulus site to the charge observed during baseline periods. This yielded a binary map showing which stimulus sites evoked significant amounts of charge (after a Bonferroni correction for multiple comparisons). We then averaged these maps together as described above.

For L4 stimulation experiments, we used widefield photostimulation through a 5x objective. We used two stimulation protocols: prolonged, 1 s ramps of linearly increasing light intensity and trains of ten pulses (1 ms duration) at 40 Hz. We stimulated at four different intensities for each protocol. Since we sometimes recorded multiple neurons in the same slice (see *Figure 2—figure supplement 1*), we fit generalized linear mixed effects models to the dose-response function of light-intensity versus evoked response (EPSC charge transfer or number of spikes), with fixed effects coefficients for the slope of this function for each cell-type and random effects slope coefficients for each slice and neuron in the dataset as well as a constant intercept term. F-tests were used to test for differences in fixed effects coefficients. For paired analysis of L4 NMCs and L5 MCs/NMCs (*Figure 2—figure supplement 1*), paired t-tests were used to test for differences in L4-evoked responses at maximum stimulus intensity.

## Two-photon CGH-based inhibitory output mapping

Laser light was generated using a 5W 1040 nm femtoTrain laser (Sepctra-Physics) and power was modulated on short time scales using a Pockels cell (Conoptics) and a high speed shutter (UniBlitz). Light was delivered to the sample using a VIVO 2- Photon workstation (3i) based on a Sutter Moveable Objective Microscope (Sutter) and the hologram was created using a Phasor 2-Photon computer-generated holography system (3i) with Slidebook software (3i) (*Figure 3—figure supplement 1a*). The holograms used for stimulation were 2D discs of diameter 15 um centered on points with 20 um spacing, making a 400 um x 400 um grid in the focal plane (*Figure 3—figure supplement 1b,c*). Stimulation consisted of 4 or 10 ms square pulses to the Pockels cell with voltages calibrated to produce 200 or 250 mW average power on sample, respectively. The choice of 4 ms at 200 mW or 10 ms at 250 mW stimulation was determined slice to slice based on opsin expression. Power for each hologram was calibrated empirically to account for power loss due to diffraction efficiency degradation away from the zero-order of the SLM. There was an inter-trial interval of 100 ms between the end of one stimulation and the start of the next stimulation. Under these conditions, SST cells spiked reliably and with high radial resolution (*Figure 3c*, *Figure 3—figure supplement 1c,d*) and moderate axial resolution (*Figure 3—figure supplement 1f,g*). Given the sparsity of SST neurons (*Figure 3b*), this level of spatial resolution provided a good tradeoff between sampling many cells with fewer targets and spiking cells with high spatial resolution. In addition, reliably evoked spikes were produced with low latency and jitter when stimulating randomly through the target grid at 10 Hz. Under these conditions, most evoked spikes occurred in the first 20 ms after the onset of stimulation (*Figure 3—figure supplement 1e,h*).

Space clamp error will inevitably affect somatic measurements of currents from distally located SST→PC synapses; however, we recorded IPSCs using a cesium-based internal solution (which included the ion channel blockers tetraethylammonium and QX-314) and performed experiments at room temperature, which ameliorate this to some extent (*Williams and Mitchell, 2008*). We also

used a holding potential of +40 mV to increase the IPSC driving force. In these experiments, internal solutions also contained 5 µM Alexa 488 hydrazide (ThermoFisher Scientific) to aid visualization with multiphoton imaging, and ~5 mM kynurenic acid Sodium salt (abcam) was added to the external ACSF to block glutamatergic activity.

To determine which locations evoked responses in the voltage-clamp recordings, first we detected IPSCs using a Bayesian modeling approach via Gibbs sampling (*Merel et al., 2016*). To obtain point estimates IPSC times from the posterior distribution over IPSC times, we binned the IPSC time samples for each trial at 1 ms resolution to create a posterior timeseries of when IPSCs were likely occurring. We then threshholded those timeseries (using *findpeaks* in MATLAB) to compute point estimates of IPSC times. Because the vast majority of evoked spikes recorded from opsin expressing SST cells occurred with short latency (*Figure 3—figure supplement 1c,f*), we estimated the evoked rate at each location from a 30 ms time window starting 5 ms after the onset of each stimulation and the background rate of IPSCs for each patched cell from the last 25 ms of all inter-trial intervals. Taking a Poisson distribution with the estimated background rate as the null distribution for all locations for each cell, we could then calculate a p-value for the hypothesis that there are no evoked IPSCs each location (i.e. there is no increase in IPSC rate). We then detected locations with evoked responses using the Benjamini-Hochberg False Detection Rate (FDR) procedure with q = 0.1 (*Benjamini and Hochberg, 1995*). We chose this relatively liberal FDR rate because any false positives will likely be thrown out after the temporal statistics are taken into account.

To determine if a location with evoked rates in both simultaneously patched cells was in fact a common input from a single source, we employed a statistical test that compares a computed synchrony statistic against a null distribution computed from resampled event time series. Specifically, the test we use employs a null distribution where all synchrony is a result of processes at timescales longer than some given duration (*Amarasingham et al., 2012*). The intuition is that the chosen duration should match the general timing of evoked IPSCs such that any synchrony under this null arises only because IPSCs across cells are being generated by two separate presynaptic SST cells stimulated on the same trials. When we reject this null, we have evidence that the synchrony is coming from a process that operates at a finer timescale than the general evoked IPSC statistics: that is, a single presynaptic SST cell is generating highly time-locked IPSCs in two postsynaptic PCs such that the across-trial-within-cell variance of IPSC times is greater than the within-trial-across-cell IPSC times. In our case, the duration of the timescale we want to test against can be estimated from both the timing statistics of evoked spiking of SST cells as wells as the peristimulus time histogram (PSTH) of IPSCs for all trials at all detected input locations across all PC input maps (*Figure 3—figure supplement 1e,h*; *Figure 3—figure supplement 2g*). Using these statistics as guidance, we chose 10 ms as the timescale for our null distribution. In detail, we first summarize the synchrony of events between two simultaneously patched cells at each location where both cells receive input. The statistic we use to quantify synchrony is the sum of the center and two flanking bins of the cross correlation of the binary event time series for simultaneously recorded cells. We then created a null distribution for this statistic at each of these locations using the event series resampling described in *Amarasingham et al. (2012)* which allowed us to estimate a p-value for each location (*Figure 3—figure supplement 2h,i,j,k*). We then detected common spatiotemporal input using these p-values and the Benjamini-Hochberg FDR procedure with q = 0.05, aggregating all tests across all paired maps. The common input probability for a simultaneously patched pair could then be computed as the total number of detected common input locations for that pair divided by the total number of unique detected input locations for the pair (i.e. the cardinality of the union of the sets of input locations for the two cells).

To align the input maps across cells, we first aligned each input map to a two-photon image of the tissue taken at the time of recording based on previous calibrations between the SLM coordinate frame (e.g. the input map frame) and the two-photon imaging frame. Next, the tissue-aligned maps were then registered via an affine transform to a confocal image of the fixed slice which had been stained with DAPI and in which the opsin expressing cells could be visualized as well as the patched cells which had been filled with biocytin. This allowed each map to be registered to each other based on the laminar borders, in particular the L4-L5 boundary.

## Biocytin staining and reconstruction

Following experiments, slices were transferred to 4% paraformaldehyde at 4° for several days. Slices were then repeatedly washed in TBS and subsequently incubated in block solution at room temperature for two hours. Next, 1:1000 streptavidin-Alexa647 conjugate was added to the solution and allowed to stain for 2 hr. Slices were then washed again and mounted/DAPI-stained on coverslips using VectaShield.

Stained neurons were imaged on a confocal microscope, along with the DAPI signal in order to identify laminar boundaries. These images allowed us to qualitatively assess whether recorded cells were L1-targeting MCs or L4-targeting NMCs. We reconstructed a subset of filled neurons, with the goal of performing a bulk quantification of how MC and NMC neurites are distributed with respect to the cortical layers (*Figure 1c*, *Figure 1—figure supplement 2e*). Since detailed reconstructions of the morphologies of these neurons have already been carried out by others (*Ma et al., 2006*; *Nigro et al., 2018*; *Wang et al., 2004*; *Silberberg and Markram, 2007*; *Xu et al., 2013*; *Tan et al., 2008*; *He et al., 2016*; *McGarry et al., 2010*), we adopted a high-throughput, semi-automated approach to perform 2D reconstruct MCs and NMCs (*Figure 1—figure supplement 2c*). We imaged neurons using a 10x air objective and used the Imaris software package to automatically trace filled neurites. Subsequently, we manually edited these traces and annotated layer boundaries. These reconstructions did not distinguish between axon and dendrite and contained small scale errors (e.g. neurites passing near each other were sometimes spuriously connected). However, comparison of semi-automated reconstructions with detailed 3D reconstructions (performed manually in Imaris, after imaging with a 60x oil immersion objective and/or a 20x air objective) showed that the semi-automated approach yielded an accurate measurement of neurite density in each layer (*Figure 1—figure supplement 2a,b*).

## Immunohistochemistry

Animals were perfused with 10 mL cold PBS followed by 15 mL 4% PFA. Brains were kept in PFA at 4 degrees for 2 hr, then washed 3 times for 15 min each in PBS while rotating. Samples cryopreserved for 24 hr in 30% sucrose in PBS at four degrees. 40 um sections were taken with a microtome. Each section washed with 0.5 mL goat blocking solution for 1 hr at four degrees, then overnight at four degrees with rat primary antibody for somatostatin in blocking solution (MAB354; Millipore; 1:100). The next day, sections washed 3 times for 15 min in PBS with 0.25% Triton X-100 (PBS-T) at room temperature while gently shaking. Sections washed with 0.5 mL in blocking solution containing goat anti-rat Alexa 647 secondary antibody for 1 hr (A21247; Life Technologies Corporation; 1:200). Sections washed 3 times for 15 min in PBS-T, then mounted on slides and coverslipped.

## Preparation for in vivo recording

Mice were anesthetized with isoflurane (2.5% vapor concentration). The scalp was removed, the fascia retracted and the skull lightly etched with a 27 gauge needle. Following application of Vetbond to the skull surface, a custom stainless steel headplate was fixed to the skull with dental cement (Metabond). Mice were allowed to recover from surgery for at least 2 d. Then mice were habituated to head-fixation on a free-spinning circular treadmill for 2–10 d. On the day of recording, mice were briefly anesthetized with isoflurane (2%), the skull over V1 was thinned and a small (<250 μm) craniotomy was opened over S1 with a fine needle.

## Tactile stimulus presentation

To stimulate the whiskers, a vertical metal bar (0.5 mm diameter) was rapidly (~50 ms) moved into the whisking field using a stepper motor with submicron precision. The bar was presented at eight different positions, evenly spanning the entire rostral-caudal axis of the whisking field, in a randomly ordered sequence. An additional ninth position that did not contact the whiskers was used to compute non-contact evoked firing rates. The horizontal distance between adjacent stimulus positions was 5.3 mm. Most mice habituated quickly to the presentation of the tactile stimulus assessed by lack of a change in whisking or running speed during stimulus presentation. Mice that did not habituate were excluded from this study. Mice were neither punished nor rewarded for any aspect of their behavior. Most mice ran consistently for hundreds of trials. Of eight stimulus positions, 3–5 contacted the principal whisker. For the X94 experiments eight mice and for the X98 experiments seven

mice were used for recordings focused on collecting data from L2/3. Control experiments were performed in three additional mice.

## Optogenetic stimulation in vivo

For optogenetic stimulation of ChR2 in vivo, we used blue light (center wavelength: 455 nm, 25 mW) from the end of a 1 mm fiber-coupled LED (Thorlabs) controlled by digital outputs (NI PCIe-6353). The fiber was placed as close to the craniotomy as possible (<3 mm). The illumination area was set to illuminate a wide area including all of S1.

Trials lasted 3 s and were separated by 1 s inter trial intervals. The motor started moving after 1000 ms and remained in the whisker field for 1500 ms. The LED switched on for 750 ms after 1500 ms in 50% of the trials. The period of light was chosen to influence the stable steady-state of the response to the whisker stimulus, and all analysis was performed during this time window.

## In vivo extracellular multielectrode electrophysiology

A 16- or 32-channel linear electrode with 25 μm spacing (NeuroNexus, A1 × 16–5 mm-25-177-A16 or A1 × 32–5 mm-25-177-A32) was guided into the brain using micromanipulators (Sensapex) and a stereomicroscope (Leica). Electrical activity was amplified and digitized at 30 kHz (Spike Gadgets) and stored on a computer hard drive. The cortical depth of each electrical contact was determined by zeroing the bottom contact to the surface of the brain. Electrodes were inserted ~25° from vertical, nearly perpendicular to the brain's surface. After some recordings a laminar probe coated with the lipophilic dye DiD was used to mark each electrode track to quantitatively assess its insertion angle and depth with *post hoc* histologic reconstructions. The laminar depth of recorded units was corrected for the insertion angle and the local curvature of the neocortex.

## Analysis of in vivo data

Spiking activity was extracted by filtering the raw signal between 800 and 7,000 Hz. Spikedetection was performed using the UltraMega Sort package. Detected spike waveforms were sorted using the MClust package (http://redishlab.neuroscience.umn.edu/MClust/MClust.html). Waveforms were first clustered automatically using KlustaKwik and then manually corrected to meet criteria for further analysis. With the exception of eight burst-firing units, included units had no more than 3% of their individual waveforms violating a refractory period of 2 ms. Individual units were classified as either fast-spiking or regular spiking using a k-means cluster analysis of spike waveform components. Since the best separation criterion was the trough-to-peak latency of the large negative-going deflection and clustering is nondeterministic, we defined all units with latencies shorter than 0.36 ms as fast-spiking and all units with latencies larger than 0.38 ms as regular-spiking. Cells with intermediate latencies were excluded from further analysis. Putative ChR2-expressing cells were identified by dramatic increases in spike rates to blue-light stimulation. The depth of each unit was assigned based on the calculated depth of the electrode on the array that exhibited its largest amplitude-sorted waveform. Layer boundaries were determined following a previously established approach (*Pluta et al., 2015*).

Firing rates were computed from counting spikes in a 750 ms window starting 500 ms after onset of the motor movement, which coincided with the onset of the LED during optogenetic suppression trials. Unless otherwise stated, we only analyzed trials when the animal was moving (at least 1 cm/s) and not accelerating or decelerating abruptly (not more than 1.5 s.d. deviation from the animal's mean running speed). Average running speed across the population was 41 ± 25 cm/s (*n* = 20 animals). Two animals were excluded because they ran fewer than 15% of total trials. To determine whether individual units were significantly modulated by optogenetic stimulation we performed an F-test on the coefficients of a Poisson generalized linear model fit to the observed firing rates of each unit.

## Tissue dissociation and FACS and 10x chromium

Six triple transgenic SST-TdT-X94 mice (age P97) were euthanized and their brains vibratome sectioned in the same manner used for acute slice experiments (described above), with the exception that slices were cut to 600 μm thickness. The somatosensory cortex was microdissected from the slices using a fine scalpel (*Pluta et al., 2015*) and allowed to recover in carbogenated 34°C sucrose

ACSF for 30 min. Following recovery, tissue was transferred to a solution of 10U/ml Papain (Worthington LK003176), dissolved in HEPES-ACSF (NaCl 120mM, KCl 5mM, MgCl2 2mM, Glucose 25mM, CaCl2 5mM, HEPES 10mM, pH 7.4 and supplemented with 1mMol solution of kynurenic acid sodium salt (Abcam 120256) and previously activated with 2.5 mM Cysteine and 2.5 mM ethylenediaminetetraacetic acid (EDTA) for 20 min at 34°C, and incubated for 25 min at 34°C under carbogen. Digestion was attenuated with 4°C Stop Solution (10% ovomuccoid inhibitor (Worthington LK003182, resuspended in EBSS to manufacturer's specifications) in HEPES-ACSF). In a volume of 3 mL, the tissue was gently triturated through a series of fire polished borosilicate glass pipettes with decreasing aperture diameters of 2mm (50 passes), 1mm (50 passes), and 0.5mm (15 passes). The resulting homogenate was passed through a 40μm cell strainer, layered over 3 mls of 20% Percoll (P4937 Sigma) in Stop Solution, and spun for 5 minutes at 400 RFC at 4°C to remove non-cellular debris. The pellet of cell bodies was resuspended in 0.2 μm-filtered Sorting Solution (HEPES-ACSF, 2% FBS) to approximately 106 cells/ml. Using a BD Influx sorter, we collected GFP+; tdTomato+ and cells into Sorting Solution (GFP+). We then collected a separate population of tdTomato+ irrespective of GFP fluorescence (tdT+) from the same batch of dissociated cells. The sorted cells were pelleted at 400 RCF at 4°C for 5 min, and resuspended in approximately 20μl Sorting Solution. 4 μl of the cell suspension was used to confirm cell concentration and cell quality by visual inspection. Separately, we performed the same procedure on two other batches of triple transgenic SST-TdT-X94 mice (age p45), but in these experiments we performed only the latter sort- collecting SST cells based on tdTomato fluorescence alone.

## Single-Cell RNA sequencing and analysis

We prepared single cell cDNA libraries from the isolated cells using the Chromium Single Cell 3' System according to manufacturer's instructions, with the sole following modification: The quantified cell suspension was directly added to the reverse transcription master mix, along with the appropriate volume of water to achieve the approximate cell capture target. We omitted the 0.04% weight/volume BSA (400 μg/ml) washing step to avoid inevitable cell loss. 2470 tdT +and 1100 GFP +cells were applied to individual channels of the Single Cell 3' Chip. The completed libraries were sequenced an Illumina HiSeq4000 to produce paired-end 100nt reads.

The libraries were processed with the 10X Genomics Cell Ranger (v. 2.0.0) pipeline, resulting in the capture of 2611 cells (232 GFP+; tdTomato +and 2379 tdTomato +irrespective of GFP fluorescence). We then used the scone (v. 1.4.0) R/Bioconductor package (*Cole, 2017*) to filter out lowly-expressed genes (fewer than 2 UMI's in fewer than 5 cells) and low-quality libraries (using the metric_sample_filter function with arguments hard_nreads = 2000, zcut = 3). This procedure resulted in a final set of 2263 cells and an average of 3160 genes detected per cell.

## Clustering of Single-Cell RNA-seq

We used the zinbwave (v. 1.3.0) Bioconductor package (*Risso et al., 2018a*) to infer a low-dimensional representation of the data (K = 10; epsilon = 1000), adjusting for batch, percentage of ribosomal genes, and total number of expressed features (computed by scater (v. 1.8.0); *McCarthy et al., 2017*). Clustering was performed on the ten-dimensional space inferred by zinbwave, using the Louvain algorithm, implemented in the FindClusters function of the Seurat package (*Butler et al., 2018*; resolution = 2). This procedure resulted in 15 clusters. We then used the clusterExperiment (v. 2.1.1) Bioconductor package (*Risso et al., 2018b*) to merge those clusters that did not show differential expression (using the function mergeClusters with arguments mergeMethod = 'adjP', cutoff = 0.05, and DEMethod = 'limma'). This procedure resulted in a final set of 13 clusters. The majority of GFP +cells fell into three merged clusters, namely m10, m1, and m12. Assignment of cluster identities was done by matching each cluster marker genes to the markers of a set of cells collected from the anterior lateral motor cortex and primary visual cortex by *Tasic et al. (2018)*. Given the absence of SST expression, we concluded that three clusters (m5, m6, and m7; total of 163 cells) were contaminants and focused on the remaining 10 clusters (2100 cells) for subsequent analysis. We then used the scmap (v. 1.2.0) Bioconductor package (*Kiselev et al., 2018*) to map the clusters onto the 2299 SST neurons identified in VISp and ALM in (*Tasic et al., 2018*).

## Identification of cluster marker genes and in situ hybridization methods

Cluster classifier gene selection: We used the clusterExperiment (v. 2.1.6) Bioconductor package (*Risso et al., 2018b*) to infer a hierarchy of the clusters and to identify the top differentially expressed genes for each cluster, using the 'OneAgainstAll' method of the getBestFeatures function, which creates contrasts to compare each cluster to the average of all the other clusters (*Supplementary file 1*). For each cluster all genes from the getBestFeatures object with a positive logFC value were examined by heat map; those that best characterized binary behavior across a given cluster definition (high expression within and low/no expression otherwise) were screened for potential quality in situ probe (good signal:noise) in the Allen Institute Brain Atlas > ISH DATA.

### ISH

Brains from three male P140 day-old mice of the genotype X94-eGFP; SST-Cre >Rosa26 LSL-tdTomato were embedded in Tissue Freezing Media on dry ice. These fresh frozen tissues were subsequently sectioned on a cryostat into 10 um coronal sections containing barrel cortex. Slides were subsequently fixed in paraformaldehyde for 15' after which they were dehydrated, protease IV digested and incubated with commercially available ACDBiotechne probes for the following genes: tdTomato (317041-C1 and C2), Calb2 (313641 C1), Hpse (412251-C1), Tacr1 (428781 C2), Timp3 (471311-C2), Pld5 (custom C2), Crh (316091 C1), Calb1 (428431 C2), and eGFP-o4 (538851-C3) according to ACDBio Fresh Frozen manual assay protocol, followed by DAPI and mounting in Vectashield. Five to seven micron optical sections were imaged at 20x using a Zeiss LSM 880 and filters for Alexa 488, Atto 550, Atto 647 and DAPI using Zen software. Z-projections and signal thresholding were performed in FIJI, using the Cell Counter plugin to record marks from manual cell counting calls. Co-expression spatial profiles are presented as a cell frequency table as triple positive as compared to double-positive (*Figure 7B*) or rastered on a common anatomy reference to bin normalized counts of cluster classifier+/tdTomato+ (as compared to total tdTomato+) by their laminar position within S1 cortex (*Figure 7C*).

## Code availability

All the code used for the analysis of the single-cell RNA-seq data is publicly available at https://github.com/drisso/x94 (*Risso, 2019*: copy archived at https://github.com/elifesciences-publications/x94).

## Acknowledgements

We are grateful to Ariel Agmon (West Virginia University) for generously providing X94 mice. We thank Connie Cepko for graciously providing viruses to pilot Cre-DOG experiments. We thank Christopher Douglas and Kirill Chesnov for technical assistance. We thank Scott Pluta and Elena Ryapolova-Webb for help with piloting experiments that did not ultimately make it into this study. We thank Ming-Chi Tsai, Alan Mardinly, Scott Pluta, Evan Lyall, Kelly Clancy, Daniel Mossing, and Ian Oldenburg for helpful comments and discussion. This work was supported by National Institute of Neurological Disorders and Stroke grant DP2NS087725-01 and the New York Stem Cell Foundation. AN is supported by the National Institute of Neurological Disorders and Stroke of the National Institutes of Health under Ruth L. Kirschstein National Research Service Award F31NS093925. BMS is supported by a Fannie and John Hertz Foundation Fellowship and an NSF Graduate Research Fellowship. LP is supported by IARPA MICRONS contract D16PC00003 and DARPA SIMPLEX contract N66001-15-C-4032. RKC, DS, DR, JN and HA were also supported by a grant from the National Institute of Mental Health to JN(U19MH114830). HA is a New York Stem Cell Foundation Robertson Investigator. DPM is supported by a National Science Foundation Graduate Research Fellowship.

# Additional information

## Funding

| Funder | Grant reference number | Author |
| --- | --- | --- |
| National Institute of Neurological Disorders and Stroke | F31NS093925 | Alexander Naka |
| Fannie and John Hertz Foundation | Fellowship | Ben Shababo |
| National Science Foundation | Graduate Research Fellowship | Ben Shababo |
| National Science Foundation | Graduate Research Fellowship | Daniel P Mossing |
| Intelligence Advanced Research Projects Activity | D16PC00003 | Liam Paninski |
| Defense Advanced Research Projects Agency | N66001-15-C-4032 | Liam Paninski |
| National Institute of Mental Health | U19MH114830 | John Ngai |
| National Institute of Neurological Disorders and Stroke | DP2NS087725-01 | Hillel Adesnik |
| New York Stem Cell Foundation | | Hillel Adesnik |

The funders had no role in study design, data collection and interpretation, or the decision to submit the work for publication.

## Author contributions

Alexander Naka, Conceptualization, Data curation, Software, Formal analysis, Validation, Investigation, Visualization, Methodology, Writing—original draft, Writing—review and editing; Julia Veit, Rebecca K Chance, Data curation, Formal analysis, Investigation, Writing—review and editing; Ben Shababo, Conceptualization, Data curation, Software, Formal analysis, Validation, Investigation, Visualization, Methodology, Writing—review and editing; Davide Risso, Data curation, Software, Formal analysis, Visualization, Methodology, Writing—review and editing; David Stafford, Benjamin Snyder, Andrew Egladyous, Desiree Chu, Investigation; Savitha Sridharan, Resources; Daniel P Mossing, Investigation, Writing—review and editing; Liam Paninski, John Ngai, Supervision, Funding acquisition, Writing—review and editing; Hillel Adesnik, Conceptualization, Supervision, Funding acquisition, Writing—original draft, Writing—review and editing

## Author ORCIDs

Alexander Naka (iD) http://orcid.org/0000-0003-0219-039X
Rebecca K Chance (iD) http://orcid.org/0000-0001-7059-6119
Davide Risso (iD) http://orcid.org/0000-0001-8508-5012
Hillel Adesnik (iD) http://orcid.org/0000-0002-3796-8643

## Ethics

Animal experimentation: All experiments were performed in accordance with the guidelines and regulations of the Animal Care and Use Committee of the University of California, Berkeley under animal protocol AUP-2014-10-6832-1.

## Decision letter and Author response

Decision letter https://doi.org/10.7554/eLife.43696.034
Author response https://doi.org/10.7554/eLife.43696.035

## Additional files

### Supplementary files

• Supplementary file 1. Table of differentially expressed genes showing one-vs-all contrasts for each transcriptomic cluster, with log fold change, p-value, and multiple comparisons adjusted p-value for each gene in each contrast.
DOI: https://doi.org/10.7554/eLife.43696.025

• Transparent reporting form
DOI: https://doi.org/10.7554/eLife.43696.026

### Data availability

All data generated or analysed during this study are included in the manuscript and supporting files. Source data files and analysis code for Figure 7 are available at https://github.com/drisso/x94 (copy archived at https://github.com/elifesciences-publications/x94).

The following previously published dataset was used:

| Author(s) | Year | Dataset title | Dataset URL | Database and Identifier |
|---|---|---|---|---|
| Tasic B, Yao Z, Graybuck LT | 2018 | Shared and distinct transcriptomic cell types across neocortical areas | https://www.ncbi.nlm.nih.gov/geo/query/acc.cgi?acc=GSE115746 | NCBI Gene Expression Omnibus, GSE115746 |

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
