## [Decision Letter]

Thank you for submitting your article "Complementary networks of cortical somatostatin interneurons enforce layer specific control" for consideration by *eLife*. Your article has been reviewed by two peer reviewers, and the evaluation has been overseen by a Reviewing Editor and Laura Colgin as the Senior Editor. The following individuals involved in review of your submission have agreed to reveal their identity: Jianing Yu (Reviewer #1); Carl CH Petersen (Reviewer #2).

The reviewers have discussed the reviews with one another and the Reviewing Editor has drafted this decision to help you prepare a revised submission.

Summary:

Naka and colleagues analyzed two types of somatostatin-expressing (SST) interneurons and their circuits in layer 5 of mouse barrel cortex. Martinotti cells send ascending axons to superficial layers and non-Martinotti (NMC) cells, which inhibit layer 4. The authors combined anatomy, multiple types of circuit analysis, and molecular methods to provide a comprehensive view of two complementary inhibitory circuits involving SST neurons. The data and analysis set a new standard for how these types of studies should be done. All reviewers were very enthusiastic about the paper and have no major criticisms.

Essential revisions:

Introduction:

In the last paragraph, the authors summarize their results as "… whereas NMC receives input from L4 and L6". Should it be "L5b/L6" (as the authors concluded in paragraph two of the Results section)? In general, the rule might be that L5 NMCs receive inputs from pyramidal cells that are excited by VPM axons (i.e. L4 and L5b/L6).

Figure 1:

The authors plot EPSC to single neurons as color maps. In Figure 1—figure supplement 4, they reported a much larger number of evoked spikes from L4 excitatory neurons (5 versus 1 for L5 neurons). A higher number of presynaptic spikes results in larger EPSCs in postsynaptic neurons, especially for those that receive facilitating inputs. Have the authors considered scaling EPSC based on evoked spike number in individual layers? The current version of the input map for non-Martinotti cells overestimates the input strength coming from L4 neurons and underestimate the input from L5b/L6.

Figure 1—figure supplement 1C and Figure 1—figure supplement 3B: Should be GOhm, not MOhm

Figure 6:

For each cell, it will be helpful to conduct a paired test to examine whether its firing rate was significantly modulated by optogenetic stimulation, and reflect the results in E and G.

In the CRE-DOG experiments, the authors inhibit L4 excitatory neurons and some L5 FS interneurons, yet the firing rate of L5 excitatory neurons seems unchanged. This seems odd given their previous paper (Pluta, 2015) where they showed that L5 FS neurons are driven by L4 excitatory neurons, and L5 E neurons are then disinhibited if L4 neurons are suppressed. This inconsistency should be discussed.

L5A differs from L5B. It would be worth making this distinction.

L4 also contains NMCs that receive input from L4 excitatory neurons and inhibit L4. Can one think of L5 NMCs as 'displaced' L4 NMCs? L4 and L5 NMCs share connectivity and gene expression patterns.

Figure 1C – it would be of interest to separately show axon and dendrite density.

Do NMCs receive excitatory input showing short-term synaptic facilitation (presumably the MCs do)?

Please discuss the current in vivo L5 data in the context of previously published L2/3 GIN & SST data (Lee et al., 2012; Gentet et al., 2012; Sachidhanandam et al., 2016; Munoz et al., 2017; Pala and Petersen, 2018).

Subsection “SST subtypes drive layer-specific effects during active sensation”: "NMC photostimulation caused little to no in the activity of the L5 RS and L6 RS populations" should probably read "NMC photostimulation caused little to no change in the activity of the L5 RS and L6 RS populations"

---

## [Author Response]

Essential revisions:Introduction:In the last paragraph, the authors summarize their results as "… whereas NMC receives input from L4 and L6". Should it be "L5b/L6" (as the authors concluded in paragraph two of the Results section)? In general, the rule might be that L5 NMCs receive inputs from pyramidal cells that are excited by VPM axons (i.e. L4 and L5b/L6).

Yes, we have corrected this. We agree that this might be a general principle that organizes NMC circuits. This idea is discussed in subsection “Functional implications of separate, layer-specific SST feedback circuits” and we have now amended the text to highlight this idea further.

Figure 1:The authors plot EPSC to single neurons as color maps. In Figure 1—figure supplement 4, they reported a much larger number of evoked spikes from L4 excitatory neurons (5 versus 1 for L5 neurons). A higher number of presynaptic spikes results in larger EPSCs in postsynaptic neurons, especially for those that receive facilitating inputs. Have the authors considered scaling EPSC based on evoked spike number in individual layers? The current version of the input map for non-Martinotti cells overestimates the input strength coming from L4 neurons and underestimate the input from L5b/L6.

This is an important consideration. Ideally, one would want to normalize by both the number and firing rates of activated neurons (see e.g. Hooks et al., 2011). Our goal in recording evoked spikes from ChR2+ excitatory cells was primarily to assess the spatial precision of our optogenetic stimulation rather than systematically measure the amount of spikes evoked by neurons in each layer. Normalizing by our recorded firing rates changes the strength of inputs dramatically – for example, the relative strength of inputs from L2/3 vs L4 are adjusted by an order of magnitude. We do not believe this accurately portrays the structure of these circuits. For example, the normalized data would suggest that L2/3 provides a stronger input to L5 NMCs than L4, but virtually no sites in L2/3 (except for those near the L4 border) actually had observable evoked EPSCs in NMC maps– suggesting that we are dramatically amplifying noise through this normalization. It is also at odds with our own unpublished observations from experiments expressing ChR2 specifically in L2/3 pyramidal cells (using the Drd3-Cre line) while recording from NMCs and MCs in L5 (see Author response image 1). Using this approach we found that L2/3 drives strong responses in L5 MCs, but produces very little responses in L5 NMCs. This contrasts quite strongly with what we saw when we optogenetically activated L4 excitatory cells (see Figure 2) and indicates that the L4 input to L5 NMCs is indeed much stronger than the L2/3 input, as our maps in Figure 1 would suggest.

To address the second point about whether short-term facilitation of glutamatergic inputs substantially affected the excitatory inputs we observed, we conducted further analysis on the excitatory currents by comparing the responses for the first, second, and third photo-stimulus in each train of brief (50 ms) light ramps we delivered (see Materials and methods). We might expect that the excitatory output of layers with high light-evoked firing rates would grow relatively stronger over time within a single trial since their synapses would facilitate more than those from layers with lower firing rates. For example, we would expect that inputs to L5 NMCs from L4 neurons (in which we observed high firing rates) would facilitate more than inputs from L5 or L6. However, we do not see any obvious change in the relative strength of inputs from different layers if we separately analyze the evoked EPSCs during different time windows in the stimulus train (Author response image 2).

**Author response image 2. respfig2:** 

Although we cannot fully explain why this is the case, one possibility is that the site-to-site variance in our maps is mainly driven by differences in the number of connected presynaptic neurons that are recruited per trial, instead of by the number of spikes or synaptic dynamics of these neurons.

In light of the above considerations, rather than normalizing our maps we have opted to include a new analysis in which we perform multiple-comparisons corrected significance testing on stimulated sites to create binary maps showing sites with/without significant evoked responses. The population averages of these maps (Figure 1 —figure supplement 3k) show a picture that is consistent with our previous interpretations, though inputs from the lower layers are relatively more prominent; this is likely because the ceiling effect produced by this procedure limits the contribution of sites with many and/or highly active presynaptic neurons to the population average. We have also amended the text in paragraph two of subsection “Two distinct sub-networks of SST neurons defined by layer-specific connectivity” to make note of some of the caveats of these experiments.

Figure 1—figure supplement 1C and Figure 1—figure supplement 3B: Should be GOhm, not MOhm

This has been corrected.

Figure 6:For each cell, it will be helpful to conduct a paired test to examine whether its firing rate was significantly modulated by optogenetic stimulation, and reflect the results in E and G.

We have conducted paired tests for each unit recorded, and find that this does reflect the results in E and G – the sign and laminar distributions of significantly modulated units are in agreement with the population level effects we show in Figure 6F and H. We have adjusted panels E and G to indicate which units are significantly modulated.

In the CRE-DOG experiments, the authors inhibit L4 excitatory neurons and some L5 FS interneurons, yet the firing rate of L5 excitatory neurons seems unchanged. This seems odd given their previous paper (Pluta, 2015) where they showed that L5 FS neurons are driven by L4 excitatory neurons, and L5 E neurons are then disinhibited if L4 neurons are suppressed. This inconsistency should be discussed.

This is an important point. There are several possible explanations, several of which we have now raised in the text. These include the much stronger reduction in L2/3 activity observed when photo-stimulating NMCs as compared to optogenetically suppressing L4 excitatory cells, which would simultaneously remove much more of the excitatory input to L5 PCs. Further experiments are needed. We have expanded our discussion of this discrepancy and separated it into its own paragraph, see paragraph three of the Discussion section.

L5A differs from L5B. It would be worth making this distinction.

We agree that this is an important distinction. Most notably, the L5 sublaminae are preferentially composed of two different classes of PCs: Type A/ thick-tufted (TT) PCs, which are often Intrinsic Bursting and are enriched in L5B, and Type B/slender-tufted (ST) PCs, which are Regular Spiking and are enriched in L5A. TT and ST cells exhibit myriad differences, including differences in their excitatory and inhibitory connectivity. We believe that this heterogeneity is unlikely to impact our conclusions about NMC circuitry, since our data indicate that NMCs are minimally connected with any kind of L5 pyramidal cell. However, it could very likely play a major role in the organization of MC circuitry. We conducted further analyses to see if our data could speak to this possibility.

We recorded from both TT and ST PCs during our paired recording experiments (see Author response image 3, panel A), and did not attempt to preferentially record from either type. Unfortunately, because the fine axons of SST cells are difficult to recover with biocytin staining, we usually filled only the SST cells to limit the amount of background biocytin signal. Consequently, we did not directly observe the dendritic morphology of most of the L5 PCs that we recorded from in these experiments. However, in paired recordings where the L5 PC was patched using potassium-based internal solution, we characterized its intrinsic properties via current injections. ST PCs are known to exhibit spike-frequency adaptation in response to prolonged current injection, whereas many TT PCs initially fire bursts or doublets of spikes before transitioning into a regular-spiking pattern that is largely non-adapting (Author response image 3, panel B) (Agmon and Connors, 1992; Chagnac-Amitai et al., 1990; Hattox and Nelson, 2007; Schubert et al., 2001). Accordingly, we reanalyzed our data and categorized our L5 PCs into putative ST PCs and putative TT PCs based on the presence or absence of burst firing, as well as on the adaptation index (see Materials and methods) of their spike trains in the last 500ms of a 1000ms current injection. We observed connectivity between MCs and both adapting (putative ST) and non-adapting (putative TT) PCs (Author response image 3, panel C). We did not observe significant differences between these groups in terms of MC→PC connectivity, nor in PC→MC connectivity (Author response image 3, panel D), though we note that splitting our data means we are likely underpowered, particularly for PC→MC connectivity.

Our new Cre-DOG optogenetic experiments also provide some insight on this question, specifically regarding MC outputs onto L5 PCs. In slice experiments, we regularly filled pyramidal neurons (see Figure 5C) while optogenetically activating MCs (in GIN-ChR2 or X98-ChR2 mice) or NMCs in X94-ChR2 mice. In cases where the dendritic morphology of L5 PCs was recovered, we classified cells as TT or ST and examined their IPSC responses to photostimulation (Author response image 3, panel E). Photostimulation in X94-ChR2 mice resulted in little response in either ST or TT cells, consistent with a lack of NMC connectivity onto L5 PCs at large. Photostimulation in GIN-ChR2 mice drove responses of similar magnitude in ST and TT cells. In X98-ChR2 mice, we were unfortunately able to recover only a relatively small fraction of PC morphologies (due to a damaged slide), but the responses of identified TT cells were similar to the responses of unidentified L5 PCs, which likely included both ST and TT cells.

**Author response image 3. respfig3:** Connectivity of MCs and NMCs with L5 PC subtypes. (**A**) Post-hoc confocal image showing biocytin fills from two paired recording experiments in connections were tested between an NMC and an ST L5 PC (left) and between an NMC and a TT L5 PC (right). (**B**) Example traces showing different spiking phenotypes in response to current injection. (**C**) Example IPSP traces from an adapting L5 PC (putatively ST) and a non-adapting L5 PC (putatively TT) showing diverging connectivity from the same MC onto both L5 PC subtypes. (**D**) Connectivity rates for MC to L5 PC inhibitory connections (left) and L5 PC to MC excitatory connections (right). (**E**) Inhibitory charge transfer observed in L5 PC subtypes in response to CRE-DOG based optogenetic stimulation of GIN, X94, and X98 cells. Each point in the swarm plots indicates the median charge transfer observed in a single L5 neuron; bold horizontal line indicates mean value for each distribution.

Because ST and TT cells are enriched in L5A and L5B, respectively, we also analyzed our in vivoCre-DOG data to see if optogenetic stimulation would differentially affect single units recorded in these sublaminae, but observed that the optogenetic effect was not significantly different between L5A and L5B for either X98-ChR2 or X94-ChR2 mice (see Author response image 4, panels A and B). We also separated our units into ‘bursty’ (if >15% of interspike intervals were 10ms or less) and ‘non-bursty’ categories but similarly did not observe any significant differences between these groups (Author response image 4, panels C and D).

**Author response image 4. respfig4:** Analysis of optogenetic effects on L5 PCs in vivo. (**A**) Plot of normalized change in firing during optogenetic stimulation versus electrode depth for regular spiking single units recorded in X94-ChR2 and X98-ChR2 mice. (**B**) Bar plots showing population effect size of optogenetic stimulation (mean ± 95% confidence interval) on regular spiking single units in L5a and L5b for X94-ChR2 (red) and X98-ChR2 (blue) experiments. (**C**) Example raster plots and autocorrelograms of a non-bursty L5 unit and a bursty L5 unit. (**D**) Swarm plot showing the normalized change in firing during optogenetic stimulation for bursty and non-bursty units in X94-ChR2 and X98-ChR2 mice. Colored points/errorbars show effect size (mean ± 95% confidence interval) in each population.

To summarize, our data indicate that NMCs exhibit little connectivity to L5 PCs, while MCs connect to both TT and ST PCs. We do not see evidence for MCs preferentially connecting to TT or ST PCs in our data, nor for optogenetic activation of MCs/NMCs differentially affecting L5A vs L5B. However, this does not mean that the heterogeneity of PCs in the L5 sublaminae does not play an important role in the organization of these circuits. Indeed, the picture emerging from recent literature suggests that parsing the heterogeneity of both PCs and MCs will be crucial to understanding these circuits. Several lines of evidence suggest that there are at least two subtypes of MCs present in L5, and likely more (Tasic et al., 2016, Ma et al., 2006, Munoz et al., 2017). The Kullander lab’s work indicates that the L5 MCs labeled by the Chrna2-Cre line preferentially target TT pyramidal cells (Hilscher et al., 2017). Since this line appears to label a subpopulation of MCs in L5 (which might correspond to the ‘T-shaped’ MCs described in Munoz et al., 2017), one possibility is that L5 contains different subtypes of MCs which preferentially connect with PC subtypes. This notion is supported by a recent study in rat frontal cortex which found strong associations between L5 MC heterogeneity and connectivity with L5 PC subtypes (Morishima et al., 2017), suggesting that different subtypes of L5 MCs might exist which preferentially connect to either ST or TT PCs.

Because our data probably do not speak definitively to specificity at this level, we have opted not to include them in the manuscript. However, we have added text making this distinction more clearly to the Discussion and Results sections.

L4 also contains NMCs that receive input from L4 excitatory neurons and inhibit L4. Can one think of L5 NMCs as 'displaced' L4 NMCs? L4 and L5 NMCs share connectivity and gene expression patterns.

We believe the evidence strongly supports the idea that L5 NMCs and L4 NMCs are the same cell-type – we previously discussed this in the supplementary text. While it is not inaccurate to describe L5 NMCs as being ‘displaced’ or ‘ectopic’ L4 NMCs, we estimate that the number of L5 NMCs in the barrel cortex is comparable to or perhaps even greater than the number of L4 NMCs. We have added new text and incorporated what was previously supplementary text discussing this into subsection “Diversity of SST cells”.

Figure 1C – it would be of interest to separately show axon and dendrite density.

We performed manual reconstructions distinguishing axon and dendrite for only 2 MCs and 2 NMCs (currently shown in Figure 1—figure supplement 1); the remainder of our reconstructions were done automatically and without distinguishing axon from dendrite (see Materials and methods), though the large majority (~90%) of the neurite density shown in Figure 1C comes from axonal arborizations. This was a deliberate decision on our part; although the automatic reconstructions are slightly less accurate than manual ones and do not make this (admittedly important) distinction between axon and dendrite, they are far less time-consuming and allowed us to collect data on a larger sample of neurons. Furthermore, during this project we became aware of another study (now published, Nigro et al., 2018) which performed detailed morphological analysis of MCs and NMCs. We have made changes to the text to make these details explicit and to call more attention to this study and others which have previously characterized the morphology of these cells.

Do NMCs receive excitatory input showing short-term synaptic facilitation (presumably the MCs do)?

Yes, excitatory synapses onto NMCs exhibit prominent short-term facilitation, similar to MCs. A quantification of this is currently shown in Figure 4—figure supplement 3B. Work from the Agmon and Connors groups has previously characterized excitatory synapses onto NMCs in some detail – we have added new citations to the Results to point the reader specifically to these studies.

Please discuss the current in vivo L5 data in the context of previously published L2/3 GIN & SST data (Lee et al., 2012; Gentet et al., 2012; Sachidhanandam et al., 2016; Munoz et al., 2017; Pala and Petersen, 2018).

We have added new text discussing these studies to the final paragraph of subsection “Functional implications of separate, layer-specific SST feedback circuits”.

Subsection “SST subtypes drive layer-specific effects during active sensation”: "NMC photostimulation caused little to no in the activity of the L5 RS and L6 RS populations" should probably read "NMC photostimulation caused little to no change in the activity of the L5 RS and L6 RS populations"

This has been corrected.